An Emerging Aerosol Climatology via Remote Sensing over Metro Manila, Philippines
Genevieve Rose Lorenzo[1,2], Avelino F. Arellano[1], Maria Obiminda Cambaliza[2,3], Christopher
Castro[1], Melliza Templonuevo Cruz[2,4], Larry Di Girolamo[5], Glenn Franco Gacal[2], Miguel
Ricardo A. Hilario[1], Nofel Lagrosas[6], Hans Jarett Ong[2], James Bernard Simpas[2,3], Sherdon Niño
Uy[2], and Armin Sorooshian[1,7]
[1]Department of Hydrology and Atmospheric Sciences, University of Arizona, Tucson, Arizona,
85721, USA
[2]Air Quality Dynamics-Instrumentation & Technology Development Laboratory, Manila
Observatory, Quezon City, 1108, Philippines
[3]Department of Physics, School of Science and Engineering, Ateneo de Manila University,
Quezon City, 1108, Philippines
[4]Institute of Environmental Science and Meteorology, University of the Philippines, Diliman,
Quezon City, 1101, Philippines
[5]Department of Atmospheric Science, University of Illinois, Urbana-Champlain, Illinois, 61801,
USA
[6]Center for Environmental Remote Sensing, Chiba University, Chiba, 263-8522, Japan
[7]Department of Chemical and Environmental Engineering, University of Arizona, Tucson,
Arizona, 85721, USA
*Correspondence to:* armin@arizona.edu
**Abstract**
Aerosol particles in Southeast Asia are challenging to characterize due to their complex life cycle
within the diverse topography and weather in the region. An emerging aerosol climatology was
established based on AERONET data (December 2009 to October 2018) for clear sky days in
Metro Manila, Philippines. Aerosol optical depth (AOD) values were highest from August to
October, partly from fine urban aerosol particles, including soot, coinciding with the burning
season in Insular Southeast Asia when smoke is often transported to Metro Manila during the
southwest monsoon. Clustering of AERONET volume size distributions (VSD) resulted in five
aerosol particle sources based on the position and magnitude of their peaks in the VSD and the
contributions of specific particle species to AOD per cluster based on MERRA-2. The clustering
showed that the majority of aerosol particles above Metro Manila were from a clean marine
source (58%), which could be related to AOD values there being relatively smaller than in other
cities in the region. The following are the other particle sources over Metro Manila: fine polluted
(20%), mixed dust (12%), urban/industrial (5%), and cloud processing (5%). Furthermore,
MERRA-2 AOD data over Southeast Asia were analyzed using empirical orthogonal functions.
Along with AOD fractional compositional contributions and wind regimes, four dominant
aerosol particle air masses emerged: two sulfate air masses from East Asia, an organic carbon
source from Indonesia, and a sulfate source from the Philippines. Knowing the local and regional
aerosol particle air masses that impact Metro Manila is useful in identifying the sources while
gaining insight on how aerosol particles are affected by long-range transport and their impact on
regional weather.

## 1. Introduction

Although Southeast Asia is one of the most rapidly developing regions in the world with a growing number of extensive research conducted (Reid et al., 2023), there remain knowledge gaps related to aerosol particles in the area (Tsay et al., 2013; Lee et al., 2018; Chen et al., 2020; Amnuaylojaroen, 2023). The region represents a complex geographic, meteorological, and hydrological environment making it challenging to understand aerosol particle characteristics, especially interactions between aerosol particles with their environment (Reid et al., 2013). The island of Luzon in the Philippines in particular is very populated and is characterized by high levels of anthropogenic emissions superimposed on natural emissions from the surrounding waters (AzadiAghdam et al., 2019) and long-range transport of emissions from areas such as Indonesia and East Asia (Braun et al., 2020; Hilario et al., 2020a; Hilario et al., 2020b; Hilario et al., 2021a). Aerosol particle lifecycle in the region is impacted by Philippine weather that is marked by two distinct monsoons, typhoons, the intertropical convergent zone, and impacts from El Niño-Southern Oscillation and Madden-Julian Oscillation (Cruz et al., 2013; Xian et al., 2013; Reid et al., 2012; Reid et al., 2015; Hilario et al., 2021b). Studying this area is informative owing to the wide dynamic range in aerosol particle and weather conditions, which are interconnected. The overlapping of large fraction of cirrus clouds with lower clouds in the area (Hong and Di Girolamo, 2020) makes space-borne remote sensing of aerosol particles very challenging (Reid et al., 2013; Lin et al., 2014). These reasons motivated the NASA Cloud, Aerosol, and Monsoon Processes Philippines Experiment (CAMP²Ex) airborne measurement campaign in 2019 to understand the interaction between tropical meteorology and aerosol particles (Di Girolamo et al., 2015; Reid et al., 2023). However, those short terms measurements cannot provide an adequate assessment of aerosol behavior across all seasons and over many years.

The NASA AErosol RObotic NETwork (AERONET) (Holben et al., 1998) is pivotal in providing broad temporal coverage of aerosol characteristics in specific locations with a column-based perspective from the ground up. Aerosol climatology studies in different regions have proved beneficial to understand temporal characteristics of aerosol particle concentrations and properties, in addition to identifying potential source regions along with interactions with clouds and rainfall (Stevens and Feingold, 2009; Li et al., 2011; Tao et al., 2012; Crosbie et al., 2014; Kumar et al., 2015; Alizadeh-Choobari and Gharaylou, 2017; Mora et al., 2017; Aldhaif et al., 2021). To our knowledge, there has not been a remote sensing-based aerosol climatology study for the Metro Manila region of Luzon, which has approximately 16 cities, a population of 12.88 million, and a high population density of 20,800 $km^{-2}$ (PSA, 2016; Alas et al., 2018).

Most of the past studies involving long-term remotely sensed aerosol particle data in Southeast Asia (Cohen, 2014; Nakata et al., 2018; Nguyen et al., 2019b) had no specific focus on the Philippines. The Philippines is considered as part of the Maritime Continent (MC), the island nations sub-region of Southeast Asia. The other Southeast Asia sub-region, Peninsular Southeast Asia (PSEA), comprises those nations within the continental Asia land mass. These two regions have separate aerosol sources and climate, where MC is dependent on the intertropical convergent zone (ITCZ) and PSEA is dependent on both the ITCZ and monsoon systems (Dong and Fu, 2015). Only the southern part of the Philippines is climatologically part of MC (Ramage, 1971), however, and northwest Philippines, where Metro Manila is located, is affected by the monsoons and tropical cyclones aside from the ITCZ (Chang et al., 2005; Yumul Jr et al., 2010; Bagtasa, 2017). These unique meteorological influences and extensive local aerosol particle

sources warrant a unique aerosol climatology over Metro Manila, one of a polluted source in a
tropical marine environment, and its effects on cloud formation in the area. Aerosol effects on
clouds in the marine environment are associated with the largest uncertainties in climate change
research (Hendrickson et al., 2021; Wall et al., 2022) and the Philippines was ranked as the 5th
country globally as most at risk to climate change and extreme weather from 1997 to 2018
(Eckstein et al., 2018). There have been several surface measurements of aerosol particles made
in Metro Manila for the past 20 years (Oanh et al., 2006; Bautista VII et al., 2014; Cruz et al.,
2019) but columnar ground-based measurements there are just beginning to be established
(Dorado et al., 2001; Ong et al., 2016; Cruz et al., 2023). The AERONET sun photometer is one
of the first long-term column-based aerosol instruments in Metro Manila and the Philippines
(Ong et al., 2016).
The goal of this study is to use multi-year AERONET data in Manila Observatory along with
other complementary datasets (MERRA-2, PERSIANN, MISR, HYSPLIT, and NAAPS) to
address the following questions: (1) what are the monthly characteristics of aerosol particles over
Metro Manila, Philippines?; (2) what are the possible sources and factors influencing the
observed characteristics?; (3) what relationships are evident between aerosol particles and cloud
characteristics?; and (4) what are the regional and local aerosol particle air masses that influence
Metro Manila?
**2. Methods**
This work relies on analysis of several datasets summarized in Table 1 and the following
subsections. The common time range used for all datasets is between January 2009 and October
110 2018.

**Table 1:** Summary of datasets over Metro Manila used in this work covering the period from
January 2009 to October 2018.

| Parameter | Data Source | Spatial Coverage | Time Coverage |
| --- | --- | --- | --- |
| Aerosol Optical Depth (500 nm) | AERONET | 14.635°N, 121.078°E | Jan 2009 - Oct 2018 |
| Asymmetry Factor (440 nm - 1020 nm) | AERONET | 14.635°N, 121.078°E | Jan 2009 - Oct 2018 |
| Extinction Angstrom Exponent (440 nm -870 nm) | AERONET | 14.635°N, 121.078°E | Jan 2009 - Oct 2018 |
| Fine Mode Fraction | AERONET | 14.635°N, 121.078°E | Jan 2009 - Oct 2018 |
| Precipitable Water | AERONET | 14.635°N, 121.078°E | Jan 2009 - Oct 2018 |
| Single Scattering Albedo (440 nm - 1020 nm) | AERONET | 14.635°N, 121.078°E | Jan 2009 - Oct 2018 |
| Refractive Index (Real and Imaginary; 440 nm - 1020 nm) | AERONET | 14.635°N, 121.078°E | Jan 2009 - Oct 2018 |
| Volume Size Distribution | AERONET | 14.635°N, 121.078°E | Jan 2009 - Oct 2018 |
| Low Cloud Fraction (MODIS) | MERRA-2 | 14.3°N - 14.8°N, 120.75°E - 121.25°E | Jan 2009 - Dec 2018 |
| Planetary Boundary Layer Height | MERRA-2 | 14.3°N - 14.8°N, 120.75°E - 121.25°E | Jan 2009 - Dec 2018 |

| | | | |
|---|---|---|---|
| Relative Humidity (975 mb) | MERRA-2 | 14.3°N - 14.8°N, 120.75°E - 121.25°E | Jan 2009 - Dec 2018 |
| Sea Level Pressure | MERRA-2 | 14.3°N - 14.8°N, 120.75°E - 121.25°E | Jan 2009 - Dec 2018 |
| Temperature (975 mb) | MERRA-2 | 14.3°N - 14.8°N, 120.75°E - 121.25°E | Jan 2009 - Dec 2018 |
| Wind (975 mb) | MERRA-2 | 14.3°N - 14.8°N, 120.75°E - 121.25°E | Jan 2009 - Dec 2018 |
| Total Extinction Aerosol Optical Depth (550 nm) | MERRA-2 | 14.3°N - 14.8°N, 120.75°E - 121.25°E | Jan 2009 - Dec 2018 |
| Sulfate, Black Carbon, Organic Carbon, Dust, and Sea Salt Extinction Aerosol Optical Depth (550 nm) | MERRA-2 | 14.3°N - 14.8°N, 120.75°E - 121.25°E | Jan 2009 - Dec 2018 |
| Precipitation | PERSIANN | 14.3°N - 14.8°N, 120.75°E - 121.25°E | Jan 2009 - Dec 2018 |

## 2.1 Datasets

2.1.1 AERONET

The central dataset used is that of sun photometer measurements and derived (inversion) parameters from the AERONET (Holben et al., 1998) site at the Manila Observatory in Quezon City, Philippines (14.64°N, 121.08°E, ~70 m. a. s. l.). Direct sunlight extinction measurements were made at nominal wavelengths of 340, 380, 440, 500, 675, 870, 940, and 1020 nm, from which aerosol optical depth (AOD) was calculated (except for 940 nm, which is for water vapor) (Eck et al., 2013). AOD is a commonly used proxy for aerosol particle loading in the air column from the ground up (Holben et al., 2001); higher AOD translates to more aerosol particle extinction in the column above a location. The extinction angstrom exponent (EAE) and the fine mode fraction (FMF) are also AERONET direct sun products that are retrieved after the application of a spectral de-convolution algorithm (O'Neill et al., 2003). For the inversion products, it is through radiative retrievals that the volume size distribution (VSD) and complex refractive index (RI) are gathered and from which single scattering albedo (SSA) and asymmetry factor (AF) are calculated. The AERONET observations were made during clear sky conditions, which has been shown (Hong and Di Girolamo, 2022) to be able to represent all sky conditions.

For the inversions, four wavelengths (440, 670, 870, and 1020 nm) of the radiometer spectral channels were chosen for diffuse radiance measurements and to avoid gas absorption (Dubovik et al., 1998). Version 3 Direct Sun and Inversion algorithms (AERONET, 2019; Giles et al., 2019) were used with the Almucantar Sky Scan Scenario to derive the following parameters with level 2.0 (automatically cloud-cleared and quality controlled datasets with pre- and post-field calibrations) data quality: column AOD (500 nm), fine mode fraction (500 nm), extinction angstrom exponent (440 – 870 nm), precipitable water (940 nm), single scattering albedo (440, 670, 870, and 1020 nm), asymmetry factor (440, 670, 870, and 1020 nm), refractive index (440, 670, 870, and 1020 nm), and VSD. The version 3 products are able to keep fine mode aerosol particle data (haze and smoke) as well as remove optically thin cirrus clouds in order to retain more aerosol particle measurements in the database (Giles et al., 2019). Cloud screening in the version 3 product improves remote sensing measurements in Southeast Asia in general, where cirrus clouds are pervasive (Reid et al., 2013). At most, a total of 29,037 direct sun and 1419 inversion AERONET daytime data points were available between January 2009 and October 2018.

2.1.2 MERRA-2

Modern Era-Retrospective Analysis for Research and Applications, Version 2 (MERRA-2: 0.5°
× 0.625° approximate resolution) meteorological and aerosol particle composition reanalysis data
(Bosilovich, 2016; Gelaro et al., 2017; Randles et al., 2017) were acquired for the area around
Manila Observatory (14.25°N – 14.75°N, 120.9375°E – 121.5625°E). The aerosol reanalysis
data includes data assimilation of AOD from the Moderate Resolution Imaging
Spectroradiometer (MODIS: Terra, 2000 to present and Aqua, 2002 to present), Advanced Very
High Resolution Radiometer (AVHRR, 1979-2002), and Multiangle Imaging SpectroRadiometer
(MISR, 2000-2014) (Buchard et al., 2017; Rizza et al., 2019). The following products were used:
M2I3NPASM Assimilated Meteorological Fields (3-hourly) for 975 mb level winds,
temperature, relative humidity, and sea level pressure; M2T1NXFLX Surface Flux Diagnostics
(1-hourly from 00:30 UTC time-averaged) 2D for planetary boundary layer height;
M2T1NXCSP COSP Satellite Simulator (1-hourly from 00:30 UTC time-averaged) for MODIS
mean low cloud fraction (cloud top pressure > 680 hPa); and M2T1NXAER Aerosol Diagnostics
(1-hourly from 00:30 UTC time-averaged) for Total AOD and speciated AOD (Sulfate, Black
Carbon (BC), Organic Carbon (OC), Dust, and Sea Salt).
MERRA-2 meteorological and aerosol particle composition monthly mean reanalysis data
(Bosilovich, 2016; Gelaro et al., 2017; Randles et al., 2017) were also acquired for a larger
region (30° × 30°), the Southeast Asia region (0°N – 30°N, 105°E – 135°E) for the period from
2009 to 2018. This is within the spatial domain of the CAMP²Ex airborne measurement
campaign which, as mentioned earlier, targets the interaction between tropical meteorology and
aerosol particles. The following datasets (0.5° latitude and 0.625° longitude resolution) were
used: MERRA-2 tavgM_2d_aer_Nx: Aerosol Assimilation (M2TMNXAER) for Total 500 nm
AOD and speciated 500 nm AOD (Sulfate, BC, OC, Dust, and Sea Salt) and MERRA-2
instM_3d_ana_Np: Analyzed Meteorological Fields (M2IMNPANA) for 1000 hPa and 725 hPa
level U and V winds. The total MERRA-2 AOD for the region (mean over 30° x 30° region) was
used along with MISR AOD data (mean over 30° x 30° region) to assess the influence of long-
range sources to the aerosol column over Manila Observatory. The monthly meteorological and
aerosol particle composition data for the region will be used for empirical orthogonal functions,
which will be described later.
2.1.3 PERSIANN
Hourly precipitation data were obtained from the Precipitation Estimation from the Remotely
Sensed Information using Artificial Neural Networks (PERSIANN) database of the Center for
Hydrometeorology and Remote Sensing (CHRS) at the University of California, Irvine (UCI).
Hourly data were accumulated for running three-day totals, which were compared to AERONET
data.  The data were averaged between the four grids that included the area of interest as well as
ensuring a similar spatial domain (14.5°N – 15.0°N, 120.75°E – 121.25°E) to the MERRA-2
dataset.
2.1.4 MISR
Monthly 500 nm AOD data (Level 3 Global Aerosol: 0.5° × 0.5° spatial resolution in the region
0.25°N – 30.25°N and 104.75°E – 134.75°E) from 2009 to 2018 are used from the Multi-angle
Imaging SpectroRadiometer (MISR), (Diner et al., 2007; Garay et al., 2018) as regional
(Southeast Asia) baseline remote sensing data to support the Manila Observatory AERONET
data. The regional (30° × 30°) MISR data was used to confirm regional sources of aerosols that
may be influencing the AOD over Metro Manila. Level 3 MISR products are global maps of
parameters available in Level 2 (measurements derived from the instrument data) products.
MISR is ideal for remote sensing in the CAMP²Ex region because it has an overpass at 10:30
AM ECT (descending mode) (when cirrus is minimal) and its retrievals have been shown to be
unimpacted by small cumulus (Zhao et al., 2009), which are typical in the region. MISR has
relatively more accurate AOD and agrees better with AERONET data compared to other satellite
products due to its multi-angle measurements (Choi et al., 2019; Kuttippurath and Raj, 2021).
The MISR sampling noise is relatively small due to the large domain and seasonal averages that
are considered in this study. MISR is also the only passive sensor that speciates aerosol particle
size and shape. All these factors led to the choice of using regional MISR data to associate long-
range sources influencing AERONET data in Manila Observatory.  Monthly mean AOD (bin 0)
were extracted for Southeast Asia (0.25°N – 30.25°N, 104.75°E – 134.75°E) within the
CAMP²Ex region. Monthly mean AOD values were then calculated for each 0.5° grid point and
then for the $30° \times 30°$ region, where the standard error in the monthly mean for the region is less
than 0.002.  MISR monthly mean time series of size, shape, and absorption speciated 550 nm
AOD and angstrom exponent in the CAMP²Ex domain (6.5°N – 22.5°N, 116.5°E – 128.5°E;
March 2000 to December 2020) are also used to support the findings from the AERONET data.
2.1.5 NAAPS
Archived maps of total and speciated optical depths and surface concentrations of sulfate, dust,
and smoke for Southeast Asia are used from the Navy Aerosol Analysis and Prediction System
(NAAPS: $1° \times 1°$ spatial resolution) (Lynch et al., 2016), and which are publicly available at
https://www.nrlmry.navy.mil/aerosol/. This reanalysis product relies on the Navy Global
Environmental Model (NAVGEM) for meteorological fields (Hogan et al., 2014). Hourly maps
were downloaded for aerosol particle events of interest based on AERONET data. These maps
help associate possible regional emission sources to extreme aerosol loading events in Manila
Observatory. Previous studies have used NAAPS data for an overview of aerosol sources in
specific regions of interest (Ross et al., 2018; Foth et al., 2019; Markowicz et al., 2021; Harenda
et al., 2022; Mims III, 2022). More recent studies show the need to improve aerosol
representation in NAAPS (Edwards et al., 2022), so we will use NAAPS qualitatively, together
with MERRA-2 compositional AOD data and back-trajectories, for an overview of aerosol
sources that may contribute to extreme events with high AOD from AERONET.
2.1.6 HYSPLIT
Back-trajectories from the National Oceanic and Atmospheric Administration's (NOAA) Hybrid
Single-Particle Lagrangian Integrated Trajectory (HYSPLIT) model (Stein et al., 2015; Rolph et
al., 2017) were used to provide support for the AERONET monthly aerosol characteristics and
the chosen case studies. Three and seven-day back-trajectories with six-hour resolution were
generated based on the NCEP/NCAR reanalysis meteorological dataset and with a resolution of
1° and a vertical wind setting of "model vertical velocity". The three-day data were used to map
the density of trajectories reaching Manila Observatory in each month from 2008 to 2019. The
seven-day data were used in the analysis of the case studies. Trajectories were computed for an
end point with an altitude of 500 m above ground level at the Manila Observatory. This altitude
represents the mixed layer based on related surface air quality studies (Crosbie et al., 2014; Mora
et al., 2017; Schlosser et al., 2017; Aldhaif et al., 2020), including a previous study for the same
area (Stahl et al., 2020).
2.1.7 NASA Worldview
Archived maps of cloud fraction (Aqua MODIS and Terra MODIS) over Metro Manila and
Southeast Asia were downloaded from NASA Worldview (https://worldview.earthdata.nasa.gov)
for events of interest based on AERONET data.
**2.2 Clustering**
Available AERONET VSD data (0.050 µm to 15.000 µm particle radius in 22 logarithmically
equidistant discrete points, 1419 data points) were clustered via k-means clustering (Lloyd,
1982). The algorithm used was k-means++ (Arthur and Vassilvitskii, 2006). The ideal number
of clusters was chosen based on relatively highest (>0.5) average silhouette value and the
presence of a cluster with a second peak in the larger accumulation mode of the VSD. The
clusters were analyzed based on their associated meteorological conditions and aerosol particle
characteristics and were classified into air mass types (Table 2) based on estimates from previous
studies (Dubovik et al., 2002; Pace et al., 2006; Kaskaoutis et al., 2007; Kaskaoutis et al., 2009;
Sorooshian et al., 2013; Kumar et al., 2014; Sharma et al., 2014; Che et al., 2015; Kumar et al.,
2015; Deep et al., 2021). The first four mentioned air mass types in Table 2 are the most general,
and four more classifications based on aerosol particle sources are included. The urban/industrial
air mass type here refers to local combustion along with long-range transported biomass burning
(Kaskaoutis et al., 2009). While these classifications are not rigid definitions of air masses, they
help in understanding the sources that contribute to aerosols in Metro Manila and in identifying
cases where certain sources are more influential than others.
**Table 2:** Summary of threshold values of aerosol optical depth (AOD), angstrom exponent (AE),
fine mode fraction (FMF), and single scattering albedo (SSA) used to identify air mass types.

| Air Mass Type | AOD | AE | FMF | SSA | Source |
|---|---|---|---|---|---|
| Clean Fine | < 0.1[a] | > 1[a] | > 0.7[a] | - | Sorooshian et al., 2013 |
| Polluted Fine | > 0.1[a] | > 1[a] | > 0.7[a] | - | Sorooshian et al., 2013 |
| Clean Coarse | < 0.1[a] | < 1[a] | < 0.3[a] | - | Sorooshian et al., 2013 |
| Polluted Coarse | > 0.1[a] | < 1[a] | < 0.3[a] | - | Sorooshian et al., 2013 |
| Clean Marine | < 0.2[b] | < 0.9[d] | - | 0.98[e] | Kaskaoutis et al., 2009 Dubovik et al., 2002 |
| Urban/Industrial | > 0.2[b] | > 1[d] | - | 0.9-0.98[e] | Kaskaoutis et al., 2009 Dubovik et al., 2002 |
| Biomass Burning | - | > 1.4[a] | - | 0.89-0.95[e] | Deep et al., 2021 Dubovik et al., 2002 |
| Desert Dust | > 0.3[c] | < 1[d] | - | 0.92-0.93[e] | Kaskaoutis et al., 2009 Deep et al., 2021 Dubovik et al., 2002 |

[a] from MODIS     [c] AOD at 400 nm     [e] SSA at 440 nm
[b] AOD at 500 nm     [d] AE at 380 nm to 870 nm


**2.3 Extreme Event Analysis**
Aerosol particle events based on the three clusters with the highest VSD concentrations were
identified to characterize different types of sources and processes impacting aerosol particle
columnar properties above Metro Manila. The three events are described below.

2.3.1   Smoke Long Range Transport
Events related to transported biomass burning/smoke were chosen from the AERONET VSD
data that were clustered as urban/industrial (with a dominant submicrometer peak) (Eck et al.,
1999) over Metro Manila. Cases with the highest black carbon contribution to total AOD from
the MERRA-2 dataset were considered. Maps from NAAPS of high smoke contributions to
AOD and surface smoke contributions in the direction of back-trajectories HYSPLIT were used
to provide support for the likely source and transport pathway for the smoke cases.

2.3.2   Dust Long Range Transport
A dust transport case over Metro Manila was identified from the AERONET VSD dust cluster
(with an enhanced coarse peak in the AERONET VSD compared to the submicrometer fraction)
(Eck et al., 1999), the highest dust contribution to AOD from the MERRA-2 dataset, and high
dust contributions to AOD from NAAPS. Surface dust concentrations from NAAPS along the
HYSPLIT back-trajectories improved the plausibility of dust for this case.

2.3.3   Cloud Processing
Cloud processing events were identified based on bimodal submicrometer VSDs (Eck et al.,
2012) and a relatively large sulfate contribution to AOD over Metro Manila from the MERRA-2
dataset, since this species is predominantly produced via cloud processing (Barth et al., 2000;
Faloona, 2009). The presence of clouds was verified qualitatively with MODIS (Aqua and Terra)
imagery from NASA Worldview in the path of air parcels reaching Metro Manila based on
HYSPLIT back-trajectories.

**2.4 Empirical Orthogonal Functions**

Regional analysis of aerosol particles in Southeast Asia and Asia in general show the prevalence
of biomass burning in the region, as well as the larger influence of anthropogenic emissions in
East Asia (Nakata et al., 2018). These large prevalent sources may overshadow other relevant but
weaker sources in the region, such as local sources. Due to the complex nature of aerosol
particles, analysis techniques such as principal component analysis and clustering along with
recent improvements in gridded datasets help detect spatial and temporal patterns that would
otherwise be difficult to make with noise interference and even weak signals (Li et al., 2013;
Sullivan et al., 2017; Plymale et al., 2021). Understanding the dominant air masses around
Southeast Asia will help in distinguishing local and transported particles that influence the
aerosol climatology in Metro Manila.

To contextualize the analysis of aerosol particle masses in Metro Manila, major regional sources
of aerosol particles in Southeast Asia were identified based on the dominant principal
components from empirical orthogonal (EOF) analysis of AOD. EOF analysis was done on the
monthly AOD data (January 2009 to December 2018) from MERRA-2 for the Southeast Asia
region for the months similar in scope to the AERONET data. EOF analysis needs a complete

dataset with no data gaps, which is not available with pure satellite retrievals like MISR; the
MERRA-2 reanalysis dataset alleviates this issue.
The monthly MERRA-2 AOD maps (0° - 30°N, 105°E – 135°E with 0.5° latitude and 0.625°
longitude resolution) (Lat: 61 rows x Lon: 49 columns) for the Southeast Asia region (presented
subsequently) were first deseasonalized. Then, the AOD anomaly per grid per year (of the 120
months) was calculated by subtracting the monthly mean AOD from each value of a given month
(Li et al., 2013).  The anomalies per grid were weighted depending on their latitude by
multiplying the anomalies by the square root of the cosine of their latitudes.
EOF, specifically singular value decomposition (SVD), analysis (Björnsson and Venegas, 1997)
was then performed. To prepare the data for the analysis, they were transformed such that the
final matrix was a 2D matrix (120 x 2989) with each row representing a year, and each column
representing a grid in the map. The matrix was analyzed for eigenvalues using SVD in Matlab,
which outputs the eigenvalue (S) and eigenvector (U: principal components and V: empirical
orthogonal functions) matrices. The eigenvalues were, by default, arranged in descending order.
Each PC time series was standardized by dividing each PC value by the standard deviation per
PC time series (120 months).
An eigenvalue spectrum was also plotted based on the variance explained by each eigenvalue
and error bars that were calculated using the North test (North et al., 1982). Then, the
unweighted AOD anomalies were regressed onto the first three standardized PCs. Each grid
therefore had a regression between 120 pairs (unweighted AOD anomalies vs standardized PCs).
From the linear regression equation, the regression coefficient per grid was calculated. Each grid
on the Southeast Asia map was colored based on the calculated regression coefficient value.
**2.5 Correlations**
The first three standardized PCs of AOD anomalies were correlated to deseasonalized
compositional AOD fractions (Sulfate, BC, OC, Dust, and Sea Salt). For each correlation, the t-
test value was calculated, and the resulting t-score was compared to a t-critical value for ~n= 100
pairs (n is the number of pairs of data, in this case 120 months) for 0.90 confidence level, which
is 1.660. Correlations that have t-values exceeding +1.660 or less than -1.660 (two-tailed test) are
significant (90% confidence).

**3   Results and Discussion**
**3.1 Meteorology and Atmospheric Circulation**
Knowledge of monthly behavior of weather in the study region helps interpretation of aerosol
particle data.  Philippine climate is influenced both by the winter northeast monsoon
(~November to April, Amihan) and the summer southwest monsoon (~May to October, Habagat)
(Coronas, 1920; Flores, 1969; Matsumoto et al., 2020). Median 3-hourly temperatures at 975 mb
per month (MERRA-2, 975 mb) (Fig. 1a) ranged from 23.2 °C in January during the winter
northeast monsoon, to 27.0 °C in May during the transition from the summer season, as defined
in (Bañares et al., 2021), to the southwest monsoon. May was also the month with the lowest
median 3-hourly relative humidity (76.6%) (MERRA-2, 975 mb) (Fig. 1b). The highest median
level of relative humidity at 975 mb for a month was in August (86.5 %) during the summer
southwest monsoon, which is also the time of the year (June to August) when rainfall peaks in
the region where the sampling station (Manila Observatory) is located (Coronas, 1920; Cruz et
al., 2013). The highest mean hourly precipitation (Fig. 1i) per month was from July (0.46 mm hr$^{-1}$)
$^{1}$) to September (0.42 mm hr$^{-1}$), while March exhibited the lowest mean hourly rainfall (0.02 mm
hr$^{-1}$).  Like relative humidity and precipitation, median precipitable water (from available
AERONET data of 513 points in August, 4015 points in February, and 5049 points in March)
(Fig. 1h) was highest in August (4.9 cm) and lowest in February and March (3.1 cm and 3.2 cm,
respectively).

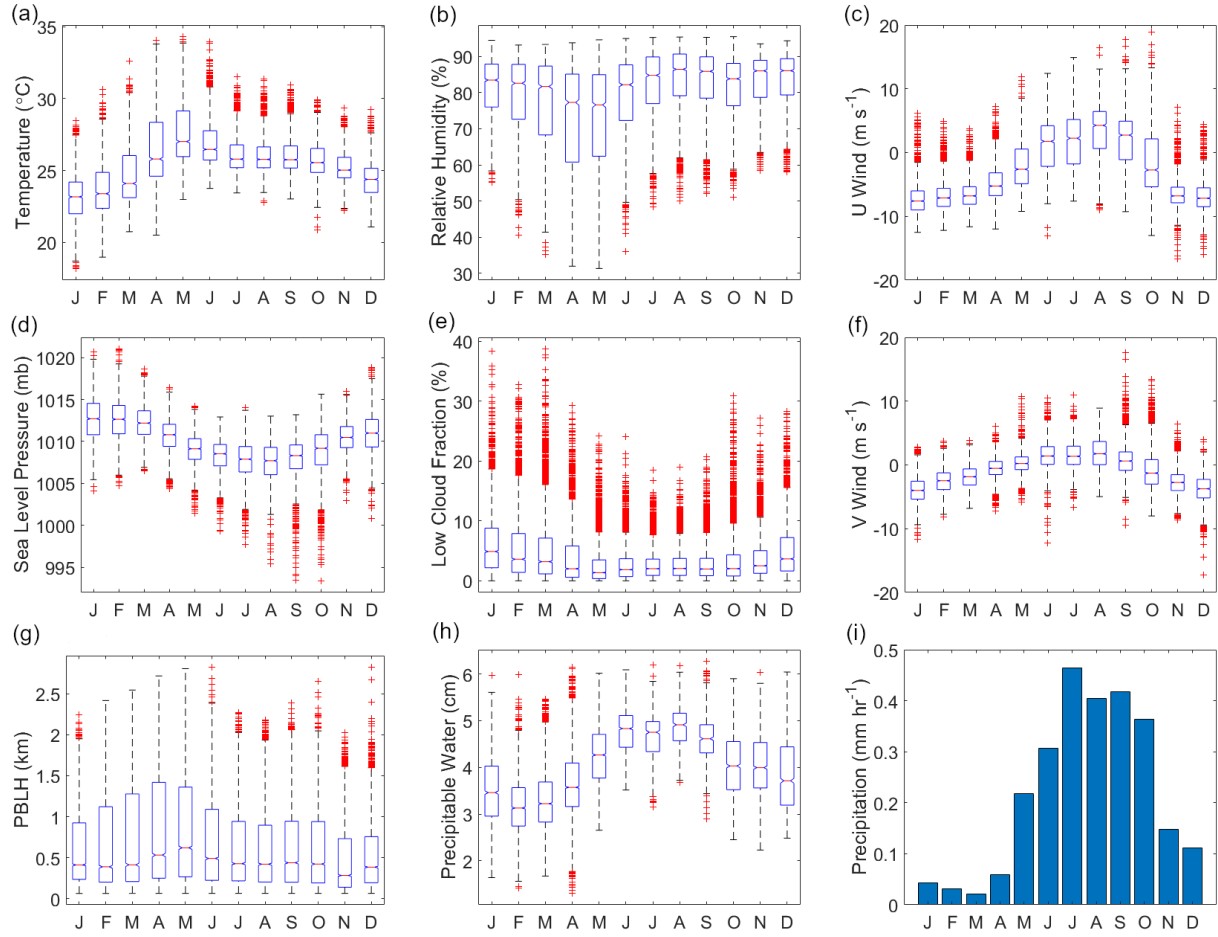



**Figure 1:** Monthly characteristics of meteorological parameters for Metro Manila, Philippines
based on data between January 2009 and October 2018. MERRA-2 parameters: (a) temperature
at 975 mb, (b) relative humidity at 975 mb, (c/f) u and v wind at 975 mb, (d) sea level pressure,
(g) planetary boundary layer height (PBLH), (e) low cloud fraction (cloud top pressure > 680
hPa); AERONET: (h) precipitable water (data counts per month Jan: 2131, Feb: 4015, Mar:
5049, Apr: 5844, May: 3448, Jun: 1696, Jul: 652, Aug: 513, Sep: 753, Oct: 1700, Nov: 2084,
Dec: 1449); PERSIANN: (i) mean hourly precipitation per month.

The lowest 3-hourly median pressures (MERRA-2) were observed (Fig. 1d) between July and
September during the southwest monsoon season (~985.2 – 985.8 mb). This is also the time
when the most number of tropical cyclones pass the island of Luzon (Wu and Choy, 2016). The
highest 3-hourly median pressures (988.1 – 990.0 mb) were during the winter northeast
monsoon.
Median winds (MERRA-2) were from the south/southwest direction from June to September
(Fig. 1c and 1f), associated with the summer southwesterly monsoon. HYSPLIT back-
trajectories show the same wind pattern (Fig. 2f to 2i). The highest median 3-hourly wind speeds
(MERRA-2) (Fig. 1c and 1f) during the southwest monsoon were recorded for August (u: 4.2 m
$s^{-1}$ and v: 1.7 m $s^{-1}$). Median winds begin to transition in October and November (to the northeast
monsoon: Amihan) (Fig. 2j and 2k) coming from the east/northeast and maintained until
February (Fig. 2b), which is towards the end of the winter northeast monsoon. There were
generally higher wind speeds and the highest median 3-hourly wind speeds of the year
(MERRA-2) (Fig. 1c and 1d) in January (u: -7.6 m $s^{-1}$ and v: -4.0 m $s^{-1}$). Median winds shifted
toward a more easterly source from March to May (transition time before the Habagat monsoon)
(Fig. 2c to 2e) accompanied by decreasing median 3-hourly wind speeds (u = -6.8 m $s^{-1}$, v = -1.9
m $s^{-1}$ to u: -2.6 m $s^{-1}$, v = 0.2 m $s^{-1}$).

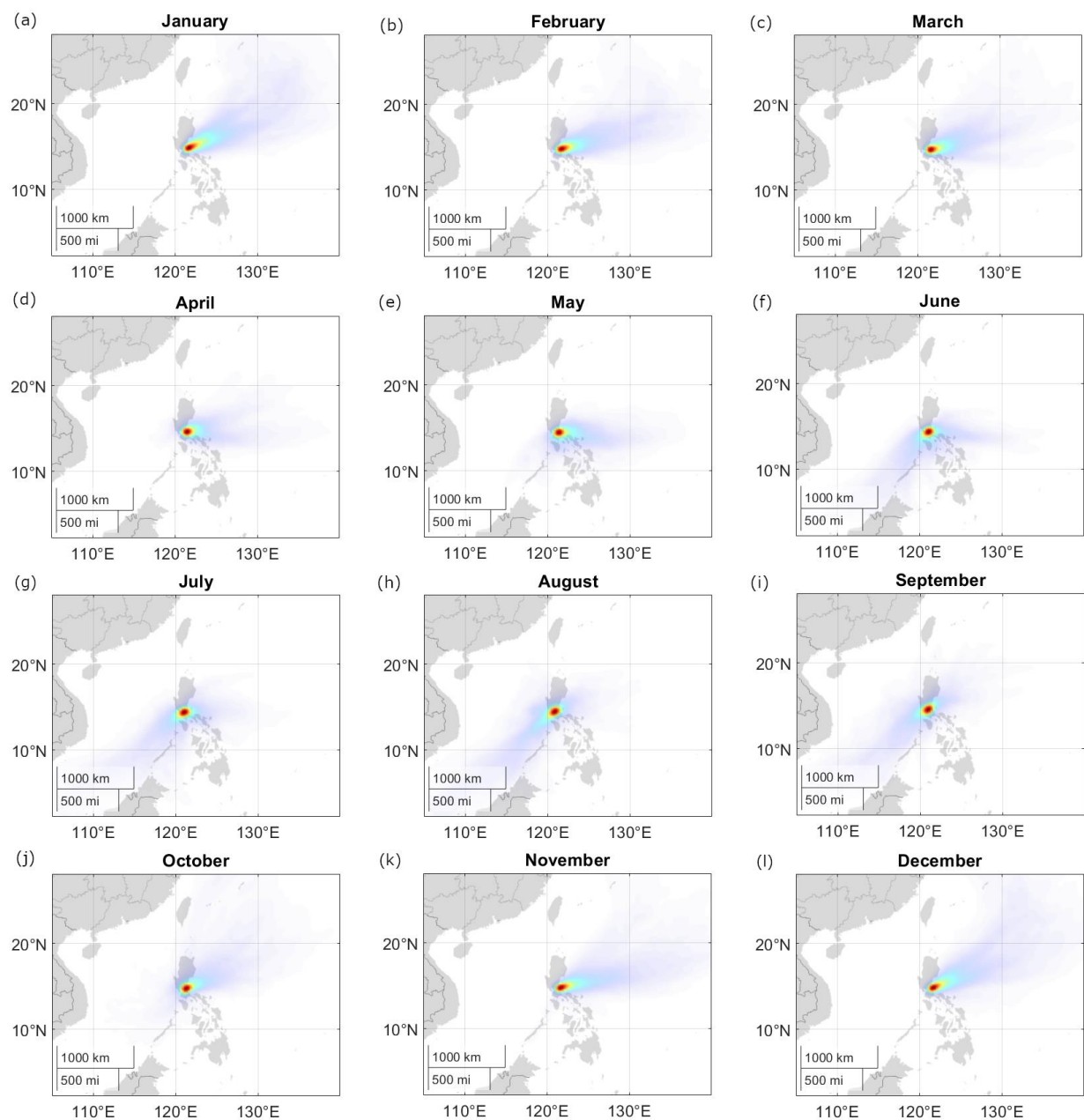

**Figure 2:** Density plots of HYSPLIT trajectories reaching Manila Observatory per month from 2009 to 2018. Red denotes areas with the greatest number of back trajectories within a 100 km radius. The colors represent density value contributions to Matlab-calculated cumulative probability distribution surfaces (100 km radius) from coordinates of three-day back trajectories of the specific months.

The transition times between the monsoons (when the wind directions shift and wind speeds change) are also the times of the highest (May, Fig. 1g, 621.2 m) and lowest (November, Fig. 1g, 279.6 m) median planetary boundary layer heights (MERRA-2). The median planetary boundary layer height was highest during the period (May) of highest temperatures, lowest relative humidity, reduced air pressure, and lowest monthly median low cloud fraction (MERRA-2) (Fig. 1e) (1.4 %). The lowest monthly median planetary boundary layer height was observed during

the period (November) when temperatures were beginning to cool and air pressure was rising.
The monthly maximum low cloud fraction was lowest in July (18.5 %) during the summer
southwest monsoon while the monthly median and monthly maximum low cloud fractions
(MERRA-2) (Fig. 1e) were highest (38.3 % max, 4.9 % median) in January during the winter
northeast monsoon.

**3.2 Aerosol Particle Characteristics**

3.2.1   Aerosol Optical Depth
Monthly median AOD (AERONET, 500 nm) (Fig. 3a) over the Manila Observatory was highest
from August (0.21) to October (0.23) around the time of the summer monsoon when winds were
coming from the southwest (Figs. 2h to 2i) (Holben et al., 2001). This is the same time of year
when biomass burning activities occur in the Indonesian region southwest of Metro Manila
(Glover and Jessup, 1998; Kiely et al., 2019; Cahyono et al., 2022). Studies have shown that
AOD in the Philippines increases during the biomass burning season in Indonesia (Nguyen et al.,
2019b; Caido et al., 2022). Regional AOD (550 nm) over the larger Southeast Asia domain from
MISR and MERRA-2 (Fig. 4) had a similarly large peak around the same time beginning in
September until October which, however, was second only in magnitude to a March peak, which
is influenced by biomass burning in Peninsular Southeast Asia (PSEA) (Gautam et al., 2013;
Hyer et al., 2013; Dong and Fu, 2015; Wang et al., 2015; Yang et al., 2022). This is consistent
with the peak in speciated AOD due to fine (radii <0.7 µm), spherical, and absorbing aerosols
that were observed by MISR from March to April (Fig. S1).  This larger peak in March,
attributed to PSEA (which is ~2000 km west of the Philippines), was not as prevalent in the
AERONET AOD data over Manila Observatory in Metro Manila due to the dominant easterly
winds in the Philippines in March (Fig. 2c) and more localized sources.

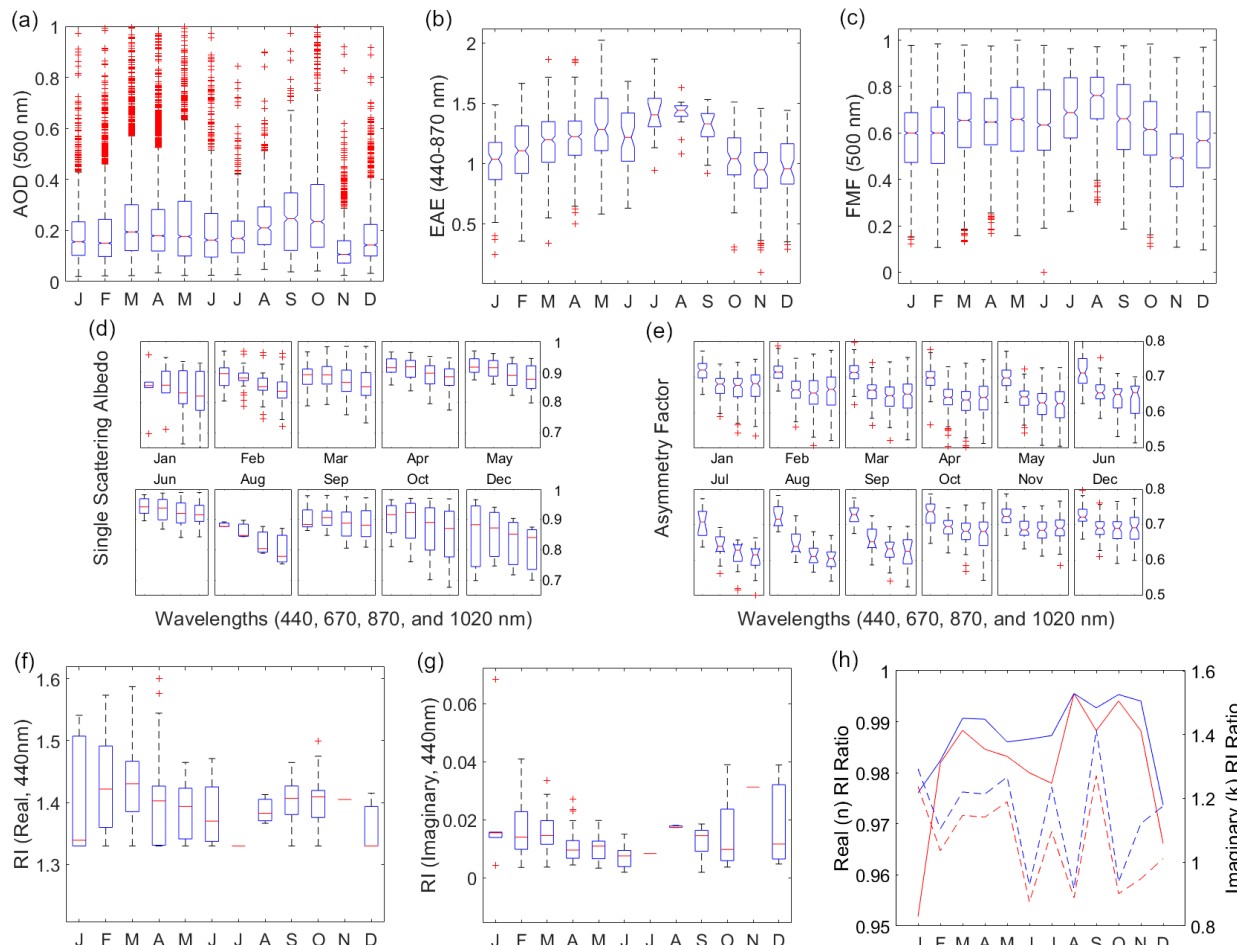

**Figure 3:** Monthly characteristics of AERONET aerosol particle parameters: (a) aerosol optical
depth (AOD at 500 nm with y-axis until 1.0 only for larger boxplot resolution) with counts (Jan:
2107, Feb: 3931, Mar: 4923, Apr: 5755, May: 3389, Jun: 1653, Jul: 637, Aug: 483, Sep: 718,
Oct: 1555, Nov: 2001, Dec: 1386), (b) extinction angstrom exponent (EAE at 440-870 nm) with
counts (Jan: 102, Feb: 248, Mar: 312, Apr: 309, May: 137, Jun: 53, Jul: 14, Aug: 18, Sep: 18,
Oct: 79, Nov: 77, Dec: 52), (c) spectral de-convolution algorithm (SDA) retrievals of fine mode
fraction (FMF at 500 nm) with the same counts as AOD, (d) single scattering albedo (SSA) from
440 nm (leftmost boxplot) to 1020 nm (rightmost boxplot) with counts (Jan: 6, Feb: 31, Mar: 62,
Apr: 50, May: 29, Jun: 8, Aug: 3, Sep: 5, Oct: 17, Dec: 3), (e) asymmetry factor (AF) from 440
nm (leftmost boxplot) to 1020 nm (rightmost boxplot) with the same counts as EAE, (f) real and
(g) imaginary refractive index (RI) values (440 nm) with the same counts as SSA, and (h)
refractive index ratios (where the blue line is the ratio of RI at 440 nm and 670 nm, the red line is
the ratio of RI at 440 nm and the average RI for the 675–1020 nm wavelengths, and the broken
lines are the imaginary refractive index ratios) for Metro Manila, Philippines based on data
between January 2009 and October 2018.

There is a notable dip in the monthly median AERONET AOD over Manila Observatory from
the peak in October to the lowest monthly median AOD (0.11) in November (Fig. 3a), just
slightly above defined background levels (<0.1) (Holben et al., 2001), when the windspeeds
were picking up and were coming from the east to northeast directions (Fig. 2k) in the direction
of the Philippine Sea and the West Pacific Ocean. This dip was also observed in the regional

(30° × 30°) AOD data (MISR and MERRA-2, Fig. 4).  This is most probably due to the decrease
in the AOD contribution from fine (radii <0.7 µm) and spherical particles based on size speciated
MISR AOD (Fig. S1). Larger and non-spherical particle contributions to AOD increase in
November in the Southeast Asia region. The MERRA-2 AOD is relatively higher than the MISR
AOD probably due to assimilation of MODIS data into MERRA-2. Studies in Asia (Xiao et al.,
2009; Qi et al., 2013; Choi et al., 2019) have observed relatively higher MODIS AOD compared
to MISR AOD.

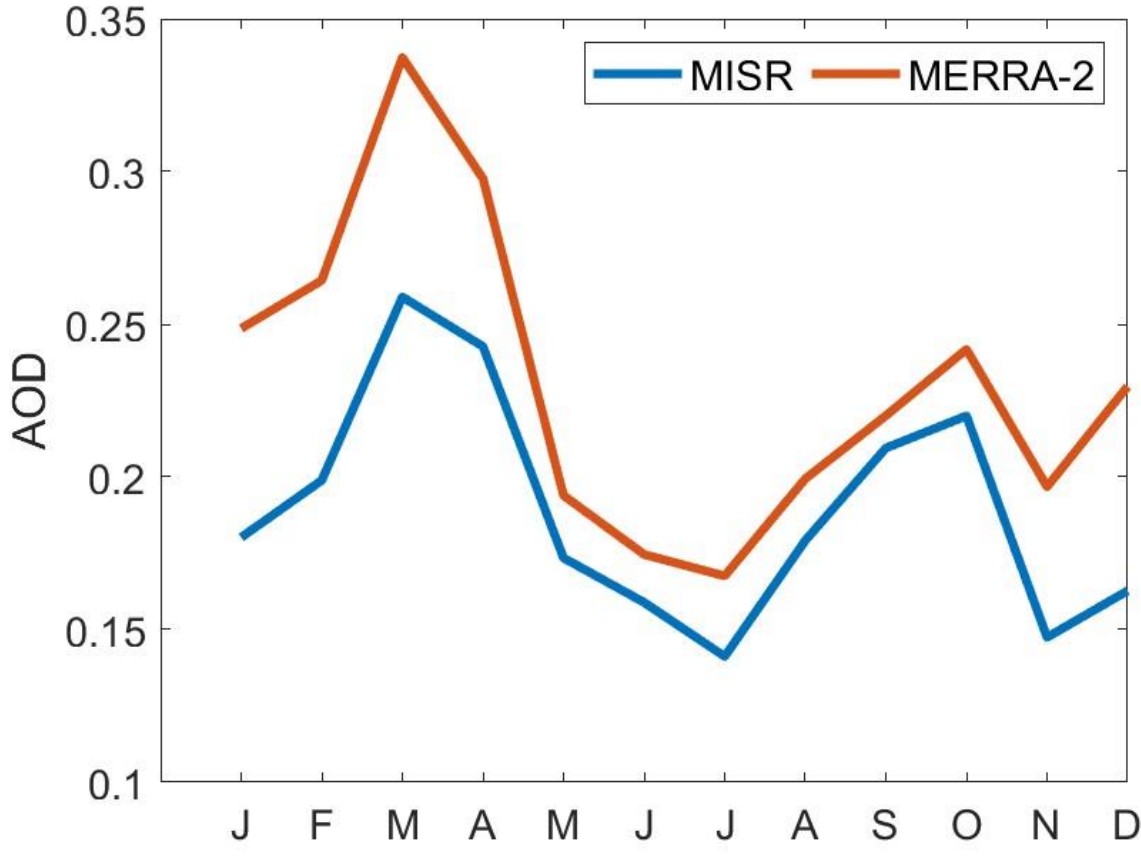


**Figure 4:** Monthly mean AOD (550 nm) in Southeast Asia (30° × 30°) from 2009 to 2018 from
MISR (blue line) and MERRA-2 (red line).
There were 338 instances (~1.2 % of the time based on the total number of 28,538 valid
AERONET AOD data points) of AOD values exceeding 1, indicative of heavy aerosol particle
loading (Huang et al., 2021). Because AOD is extrinsic (it depends on mass), AOD describes
total aerosol particle loading and we examine other aerosol particle parameters from AERONET
to make more informed inferences about size and composition.
3.2.2   Extinction Angstrom Exponent and Fine Mode Fraction
The extinction angstrom exponent (EAE) relates the extinction of light at specific wavelengths
and is indicative of aerosol particle size (Ångström, 1929). The EAE is usually greater for
smaller particles (~4 for very small particles that undergo Rayleigh scattering, > 2 for small
particles, < 1 for large particles like sea salt and dust, and 0 for particles as large as cloud drops)
(Schuster et al., 2006; Bergstrom et al., 2007). The highest monthly median EAE (Fig. 3b) from
2009 to 2018 over the Manila Observatory was observed from July (~1.4) to September (~1.3),
during the southwest monsoon. This period is associated with the biomass burning southwest of
the Philippines (Oanh et al., 2018; Stahl et al., 2021; Crosbie et al., 2022). The median (per
month) EAE ranged from ~0.9 in November to ~1.4 in August, a range which is within the
values from previous studies collected from mixed sites and urban/industrial areas with both fine
and coarse particles (Eck et al., 2005; Giles et al., 2012). The high EAE over Manila Observatory
from July to September is consistent with the regional (30° latitude x 30°longitude) MISR data
that shows increased AOD from fine, spherical, and absorptive particles (Fig. S1) in Southeast
Asia during the same months. This suggests that the high EAE observed at the Manila
Observatory during these months is not necessarily from local sources.
EAE increases with AOD (Fig. S2), which means that the greater particle loading is contributed
by smaller particles (Smirnov et al., 2002). Of the high loading cases (AOD >1) over Manila
Observatory, the EAE values were mostly greater than 0.8 indicating fine mode particles (Che et
al., 2015). The EAE values in August were the highest compared to other months including
having the highest minimum value of any month (0.71) (Fig. S2), due to smaller particles (~EAE
>1 for fine particles, Table 2). The lowest EAE values (0.08) and thus the largest particles were
observed in December, which again may be regional in nature with MISR EAE also lowest
during this time with increased AOD from larger and non-spherical particles (Fig. S1).
The fine mode fraction (FMF) describes the prevalence of fine mode particles in the column of
air above the surface. The fine mode fraction (Fig. 3c) from 2009 to 2018 was highest in August
(monthly median of 0.75) and lowest in November (monthly median of 0.45). This is consistent
with the EAE values discussed earlier with the prevalence of smaller particles in August and
larger particles in November. In August (Fig. 2h) the southwest monsoon is known to coincide
with the transporting of fine smoke particles to Luzon. In November (Fig. 2k), the prevalent
winds may have already shifted to easterly (Matsumoto et al., 2020) implying more marine-
related sources associated with coarser particles.
3.2.3  Single Scattering Albedo
The single scattering albedo (SSA) is the most important aerosol particle parameter determining
whether aerosol particles will have a warming or cooling effect (Reid et al., 1998). SSA is the
ratio of the scattering coefficient to the total extinction (scattering and absorption) coefficient
(Bohren and Clothiaux, 2006) of aerosol particles. Higher SSAs are related to more reflective
aerosol particles while more absorbing aerosol particles will have lower SSA values; values
range from 1 (reflective) to 0 (absorbing). Monthly median SSA values were largest in June
(0.94 at 440 nm), suggesting the presence of more reflective aerosol particles, and smallest in
August (0.88 at 440 nm and 0.78 at 1020 nm) suggesting more absorptive particles that are
similar in range to the SSA of biomass burning particles (Table 2). August is when biomass
burning is prevalent to the southwest of the Philippines and associated with soot particles that are
absorptive.
The sensitivity of SSA to different wavelengths depends on the type of aerosol particles present.
More specifically, aerosol particle size and refractive index (which is related to aerosol particle
composition) both affect the SSA (Dubovik and King, 2000; Bergstrom et al., 2007; Moosmüller
and Sorensen, 2018).  For dust-type particles, SSA increases with wavelength because of lower
dust absorption in the higher visible to infrared wavelengths (Dubovik et al., 2002), while for
urban particles (including black carbon), which absorb light at longer wavelengths, SSA
decreases with wavelength (Reid et al., 1998; Bergstrom et al., 2002). The presence of organic
carbon may affect this spectral dependence; however, because organic particles absorb in the
UV, this lowers SSA at wavelengths shorter than 440 nm (Kirchstetter et al., 2004). Monthly
median SSA generally decreased with increasing wavelength for all months with available data
(Fig. 3d) presumably due to the influence of more urban particles in contrast to dust.
Noteworthy though are the monsoon transition months of April, September, and October (Fig.
3d), which had increased SSA from 440 nm to 670 nm, possibly from organics along with black
carbon due to transported smoke. The back-trajectories for these months (Figs. 2d, 2i, and 2j)
suggest sources from the northeast that are closer to Luzon during these months compared to
other months. This indicates the possibility of more local sources.  Increasing the certainty of
sources associated with aerosol particles necessitates looking at other available aerosol particle
parameters, discussed subsequently.
3.2.4   Asymmetry Factor
The asymmetry factor quantifies the direction of scattering of light due to aerosol particles, with
values ranging from -1 (back scatter) to 0 (uniform scattering) to 1 (forward scatter). It is
important in modeling climate forcing because it affects the vertical distribution of the radiation
in the atmosphere (Kudo et al., 2016; Zhao et al., 2018). The asymmetry factor is dependent on
particle size, shape, and composition and the value of 0.7 is used in radiative models (Pandolfi et
al., 2018).
Lower asymmetry factors are related to smaller particles (at constant AOD) (Bi et al., 2014).
Measured values due to biomass burning, for example, are 0.54 (550 nm) in Brazil (Ross et al.,
1998) and 0.45 – 0.53 (550 nm and including dust) over central India (Jose et al., 2016). There
have been relatively higher values observed in western, central, and eastern Europe (0.57 – 0.61
at 520 – 550 nm) (Pandolfi et al., 2018) and the U.S. East Coast (0.7 at 550 nm)  (Hartley and
Hobbs, 2001).  In Norway, the asymmetry factor for background summer conditions was 0.62
and was higher in the springtime at 0.81 (862 nm) during Arctic haze events (Herber et al.,
2002). Highest values are associated with dust such as those measured in the Sahara being 0.72 –
0.73 (500 nm) (Formenti et al., 2000). Over Metro Manila, the asymmetry factors from the
AERONET data at the 675, 870, and 1020 nm were similar across months (Fig. 3e). The monthly
median asymmetry factors at 440 nm ranged from 0.70 (April and May) to 0.74 (October), while
for 670, 870, and 1020 nm the monthly median asymmetry factors were smaller and ranged from
0.62 – 0.69. These values were closely related to those observed over the U.S. East Coast as
mentioned earlier, perhaps due to the proximity of the location to the coast (10 km east of Manila
Bay and 100 km west of the Philippine Sea) as well as its location in Manila, which is a large
local source due mostly to vehicles (Cruz et al., 2019).
The monthly median asymmetry factor in Metro Manila was greatest towards the end of the year
(October to December) for all the wavelengths, suggesting larger particles when winds (Figs. 2j
to 2l) come from the Philippine Sea in the northeast. It was in March and April that the monthly
median asymmetry factor was minimal for 440 nm and in August for 670, 870, and 1020 nm.
These were the times when aerosol particles were smallest. March to April represents the driest
time of the year in Manila (Fig. 1b and 1h) perhaps preventing particle growth and where the
local sources may be dominating, even as back-trajectories (Fig. 2c and 2d) extend all the way
from the Philippine Sea to the east.  This is corroborated by results from other studies showing
that the asymmetry factor seems to be enhanced by relative humidity (Zhao et al., 2018). The
unexpected low asymmetry factor values in August, however, are probably because of the source
of the particles.  August had the highest relative humidity and precipitable water (Fig. 1b and 1h)
but is also when the back-trajectories (Fig. 2h) were from the southwest, possibly affected by the
Indonesia fires, which could have transported more non-hygroscopic fine particles.
Fine particles have been observed to exhibit decreasing asymmetry factors with increasing
wavelength (Bergstrom et al., 2003). This trend is observed in all the months for the monthly
median asymmetry factors (Fig. 3e) suggesting the predominance of smaller aerosol particles.
The greatest decrease in the asymmetry factor (all wavelengths) was in August, consistent with
the lowest observed values of the year (670, 870, and 1020 nm). Transported biomass burning
particles are the probable dominant particles during this time.  They are usually composed of
hygroscopic inorganics, non-hygroscopic soot, and relative non-hygroscopic organic fractions
(Petters et al., 2009). Knowing the composition of biomass burning particles over the study
region will help in the understanding of hygroscopicity and its impacts on radiation.
3.2.5   Refractive Index
Refractive index is an intrinsic parameter as it does not depend on the mass or the size of
particles, and thus can be used to infer aerosol particle composition (Schuster et al., 2016). For
the case of the AERONET data, which include refractive index values that are insensitive to
coarse particles (Sinyuk et al., 2020), the focus of the discussion will be for fine mode particles
and may be limited when coarse particles are involved.   Refractive index measurements are
complex since they include real and imaginary parts related to light scattering and absorption,
respectively. All aerosol particles scatter light but only certain types absorb light significantly.
The most prominent particle absorbers in the atmosphere are soot carbon, brown carbon (organic
carbon that absorbs light), and free iron from dust (hematite and goethite in the ultraviolet to
mid-visible) (Schuster et al., 2016). For this study, we examine refractive index values at 440 nm
wavelength. Pure sources of soot carbon have the highest real refractive index values (~1.85) as
well as the highest imaginary refractive index (~0.71), both independent of wavelength (Koven
and Fung, 2006; Van Beelen et al., 2014). Brown carbon and dust have relatively lower real
refractive index values at 440 nm (~1.57 and ~1.54) and imaginary refractive index values
(~0.063 and ~0.008) that decrease with increasing wavelength (Xie et al., 2017).
In this study the range of the monthly median real refractive index values (440 nm) was from
1.33 (December and January) to 1.43 (March) (Fig. 3f). Water uptake by aerosol particles
decreases the real refractive index values (Xie et al., 2017) and thus the lowered real refractive
indices over the Manila Observatory can be due to the presence of more water in the atmosphere
in general and/or the increased presence of more hygroscopic particles. December and January
are not necessarily the months that have the highest moisture content, but they are months when
back-trajectories reaching the column over the Manila Observatory are from the Philippine Sea
to the northeast presumably transporting hygroscopic particles. As reported in previous sections,
relatively larger particles are observed around this time of the year and thus sea salt can be an
important contributor. The greatest change in the monthly median real refractive index with
increasing wavelength also was observed in December (Fig. 3h), possibly due the increased
fractional contribution of constituents other than soot carbon (because the real refractive index of
soot carbon is invariant with wavelength). Noteworthy as well is the month of August (Fig. 3f),
which has the smallest range of real refractive index values, possibly indicating a more
homogenous aerosol particle source compared to other months. August is the month with the
highest relative humidity (Fig. 1b) as well as highest precipitable water (Fig. 1h), while this is
also the month when long-range biomass burning emissions are observed to be highest, and
when the real refractive index values would otherwise be expected to be highest.
Water content seems to play a significant role in the real refractive index values in Manila.
March, when the monthly median real refractive index values are highest (Fig. 3f), is when
precipitable water vapor (Fig. 1h) is among the lowest in the year. The months around March are
also when maximum real refractive indices (1.57 in February, 1.59 in March, and 1.60 in April)
were observed (Fig. 3f). March was when there was a relatively small change in real refractive
index value with wavelength perhaps related to greater soot carbon fractions during this time,
due possibly to the contribution of biomass burning from Peninsular Southeast Asia (Shen et al.,
2014). Looking more closely at the imaginary refractive index values will help elucidate this
issue.
Monthly median imaginary refractive index values (440 nm) ranged from 0.007 in June to 0.015
in September and December (Fig. 3g). These are low compared to those of the pure soot carbon
mentioned earlier because of the mixed nature of the sampling site with contributions from
brown carbon and dust. The highest imaginary refractive index values in September and
December suggest the greatest fractional contribution of soot because the highest imaginary
refractive index values are associated with soot. These are also similar in magnitude to biomass
burning particles in the Amazon (0.013) (Guyon et al., 2003). The key distinction between soot
carbon and other major absorbers (brown carbon and dust) is that its imaginary refractive index
is invariant with wavelength. Both brown carbon and dust exhibit a decrease in the imaginary
refractive index with increasing wavelength (Xie et al., 2017). The ratios of imaginary refractive
index values (440 nm to average of 670–1020 nm) (Fig. 3h) show a relative invariance with
wavelength (ranging from 0.88 to 1.4), which indicates the dominance of soot as the major
absorber in the region (Eck et al., 2003). While observed wavelength invariance points to high
soot contributions, the size of the particles can help distinguish between brown carbon, which
reside mainly in the fine mode, and dust sources, which yield more coarse particles (Schuster et
al., 2016). September is during the southwest monsoon, which is when, as noted in the earlier
sections, fine particles were most prevalent. This is also the time when the imaginary refractive
index varied most with wavelength (1.4 ratio of the imaginary refractive index at 440 nm and the
imaginary refractive index average for 670 nm to 1020 nm in Fig. 3h) possibly with greater
absolute contributions from brown carbon, even with the highest soot carbon fractional
contributions. Brown carbon has been observed both from primary and aged aerosol particle
emissions from biomass burning (Saleh et al., 2013). As noted earlier, December also had the
highest imaginary refractive index values as well as relatively coarser particles, possibly due to
larger dust absolute contributions even with the highest soot carbon fraction contributions. The
lowest monthly median imaginary refractive index values in June, on the other hand, when fine
mode particles prevail suggest highest fractional contributions of brown carbon relative to other
months (Fig. 3h).
3.2.6 Volume Size Distributions
The volume size distribution (VSD) is another way to be able to more deeply characterize
aerosol particles, specifically related to their effect on climate, weather, and clouds (Haywood
and Boucher, 2000; Feingold, 2003). In the Manila Observatory dataset, there was a bi-modal
VSD for the entire dataset (Fig. 5a). The fine mode median values peaked in the accumulation
mode at 0.148 µm particle radius while the coarse mode median values peaked at 3.857 µm (Fig.
5a and Table S1). The median coarse mode amplitudes and volume concentrations were higher
than the fine mode amplitudes and volume concentrations for most of the year (DJF, MAM, and
SON, Fig. 5b and Table S1), except during the southwest monsoon (JJA) when the fine mode
amplitude and volume concentration was higher. This is consistent with observations earlier of
fine mode prevalence during the southwest monsoon. Median VSD amplitudes (Fig. 5c) were
greater in the afternoon, with higher peaks and volume concentrations for both the fine and
coarse modes, compared to the morning. There was a slightly larger coarse median amplitude
and volume concentration, compared to the accumulation mode median amplitude and volume
concentration, for both the morning and afternoon size distributions. While the VSDs confirm
several observations based on the analysis of the aerosol particle parameters presented earlier,
not much further information is gained especially regarding chemical composition. Size
distributions are a result of contributions from multiple sources, and thus being able to
discriminate the sources based on their characteristic size distributions will help identify relevant
sources.

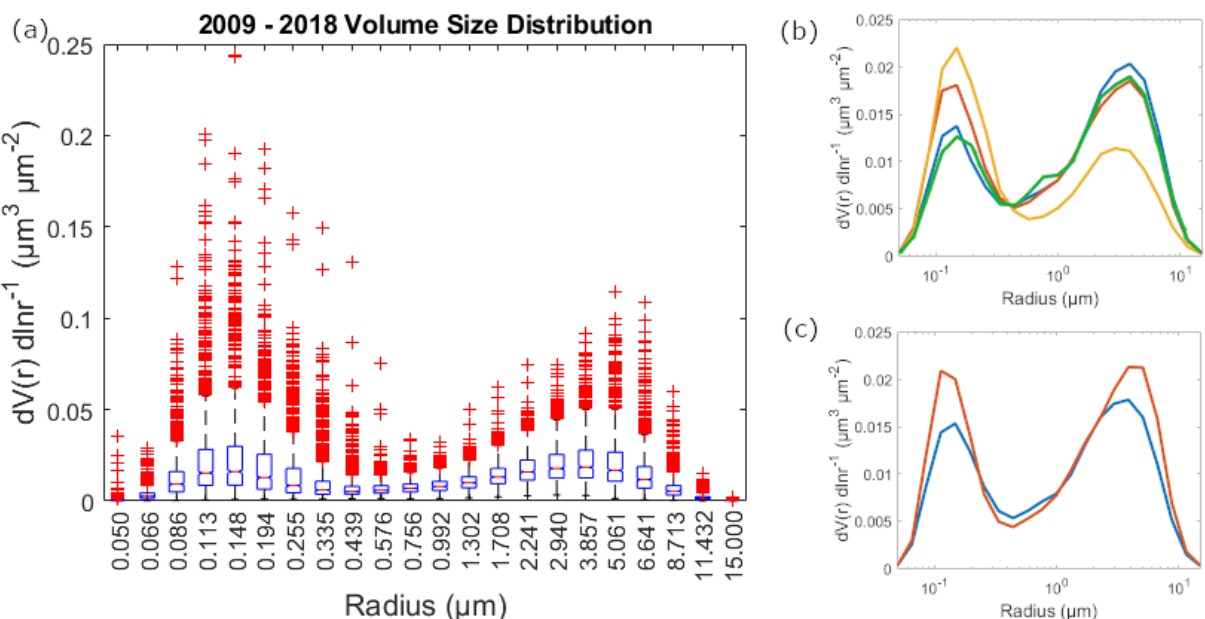


**Figure 5:** (a) VSD results derived from AERONET measurements at Metro Manila between
January 2009 and October 2018. Median VSDs over the study period based on (b) season (blue:
DJF, red: MAM, orange: JJA, green: SON) and (c) time of day (blue: AM, red: PM).

**3.3 Clusters**
3.3.1 VSD Cluster Profiles
Five clusters were identified to best represent the VSD (Fig. 6a). The average of the VSDs in
each cluster varied depending on the height of the peaks in the accumulation mode and the
coarse mode. In Metro Manila, the accumulation mode is associated with aged aerosol particles
and combustion (Cruz et al., 2019). The majority of the data (830 count out of 1419 total VSD
profiles) were clustered together in a profile (cluster 1) that had relatively low average
magnitudes of volume concentration for both the accumulation (0.01 $\mu m^3$ $\mu m^{-2}$) and coarse (0.02
$\mu m^3$ $\mu m^{-2}$) modes, with the volume concentration magnitude of the coarse mode peak slightly
higher than the volume concentration magnitude of the accumulation mode peak. The next
prevalent cluster profile (284 counts, cluster 2) had an average fine mode peak for the volume
concentration (0.04 $\mu m^3$ $\mu m^{-2}$) which was more than twice as much than the previous profile but
with a similar coarse mode peak for the volume concentration (0.02 $\mu m^3$ $\mu m^{-2}$). The average
coarse mode peak for the volume concentration (0.04 $\mu m^3$ $\mu m^{-2}$) was the highest (compared to
the four other cluster profiles) for the third prevalent cluster profile (166 counts, cluster 3);
cluster 3 also had a slightly shifted volume concentration peak in the coarse mode to a higher
radius (5.06 $\mu m$) compared to other clusters. The coarse mode dominated this VSD compared to
other profiles (lower magnitude for the accumulation mode peak for the volume concentration,
0.02 $\mu m^3$ $\mu m^{-2}$). The two remaining cluster profiles exhibited high average magnitudes of
volume concentration in both the accumulation and coarse modes. The fourth prevalent cluster
profile (74 counts, cluster 4) had the highest average absolute magnitude for the volume
concentration in the accumulation mode (0.11 $\mu m^3$ $\mu m^{-2}$), while the fifth prevalent cluster profile
(65 counts, cluster 5) had a slightly smaller accumulation mode peak for the volume
concentration (0.07 $\mu m^3$ $\mu m^{-2}$) that was shifted to a slightly higher radius (0.19 $\mu m$ compared to
0.15 $\mu m$). Both clusters 4 and 5 had similar average coarse mode peak volume concentration
magnitudes (0.04 $\mu m^3$ $\mu m^{-2}$).

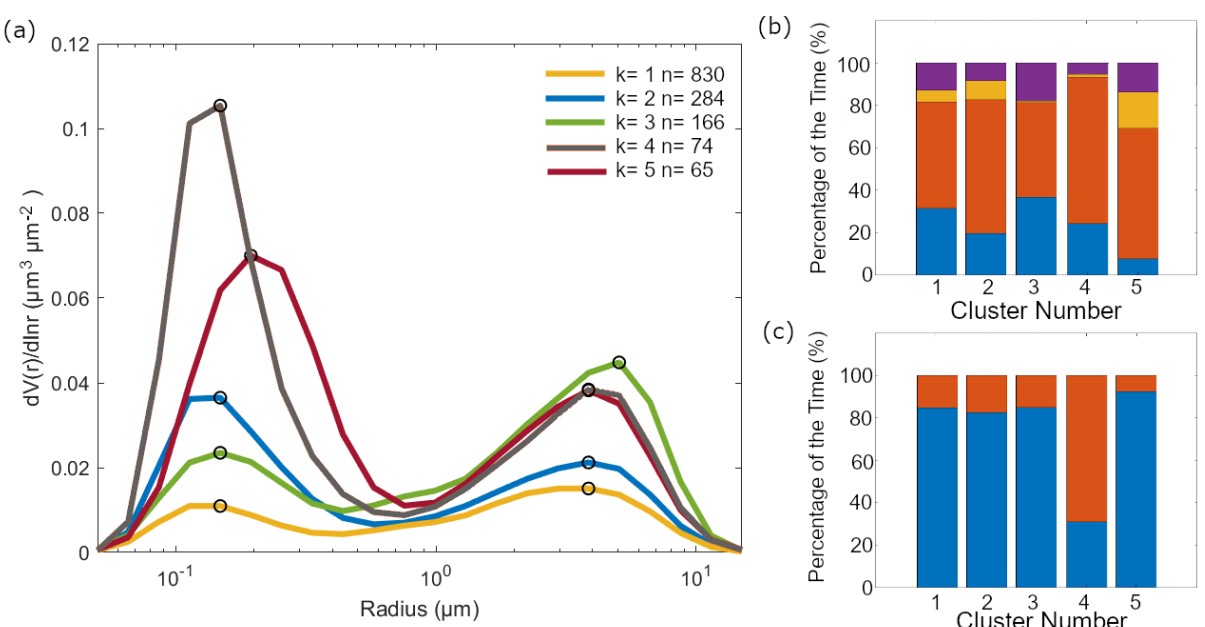

**Figure 6:** (a) Cluster analysis of VSD data yielding five characteristic and averaged VSDs with
the number of points per cluster shown in the legend. The black circles on the curves show the
peak locations in the submicrometer (<1 µm) and coarse (≥1 µm) modes. The relative abundance
of each cluster is shown for different (b) seasons (blue: DJF, red: MAM, orange: JJA, violet:
SON) and (c) times of day (blue: AM, red: PM).
The clusters were distributed across seasons (Fig. 6b), with clusters 1 and 2 being the most
evenly distributed among the clusters. Cluster 3, which had the highest coarse mode peak, had
the greatest contribution from September to November compared to other clusters. Cluster 4,
which had the highest accumulated mode peak compared to other clusters, had the greatest
contribution from March to May as well as to afternoon VSDs compared to other clusters (Fig.
6b and 6c). Relative contributions of VSDs from June to August were highest for cluster 5,
which had the shifted accumulated mode peak.
Median total (AERONET) AOD values (Fig. 7b) were lowest (0.12) for cluster 1, though it had
the second highest sea salt fractional contributions (31%) (Fig. 7a) to total AOD (MERRA-2)
among all the clusters. Cluster 2 had relatively mid-range median total AOD values (0.27) that,
along with clusters 4 and 5, were dominated by sulfate and organic carbon (46% and 20%).
Cluster 3 had similar, but slightly lower median total AOD (0.25) compared to cluster 2. Cluster
3 was distinct because it had the largest total (0.04) and fractional contribution (37%) from sea
salt among all clusters. Clusters 4 and 5 had the highest median total AOD values (0.47 and
0.56), with cluster 5 having the highest absolute and fractional sulfate contributions (0.14 and
64%) among the clusters.  Integrating the above results with their corresponding aerosol particle
properties can help associate the clusters to air masses.

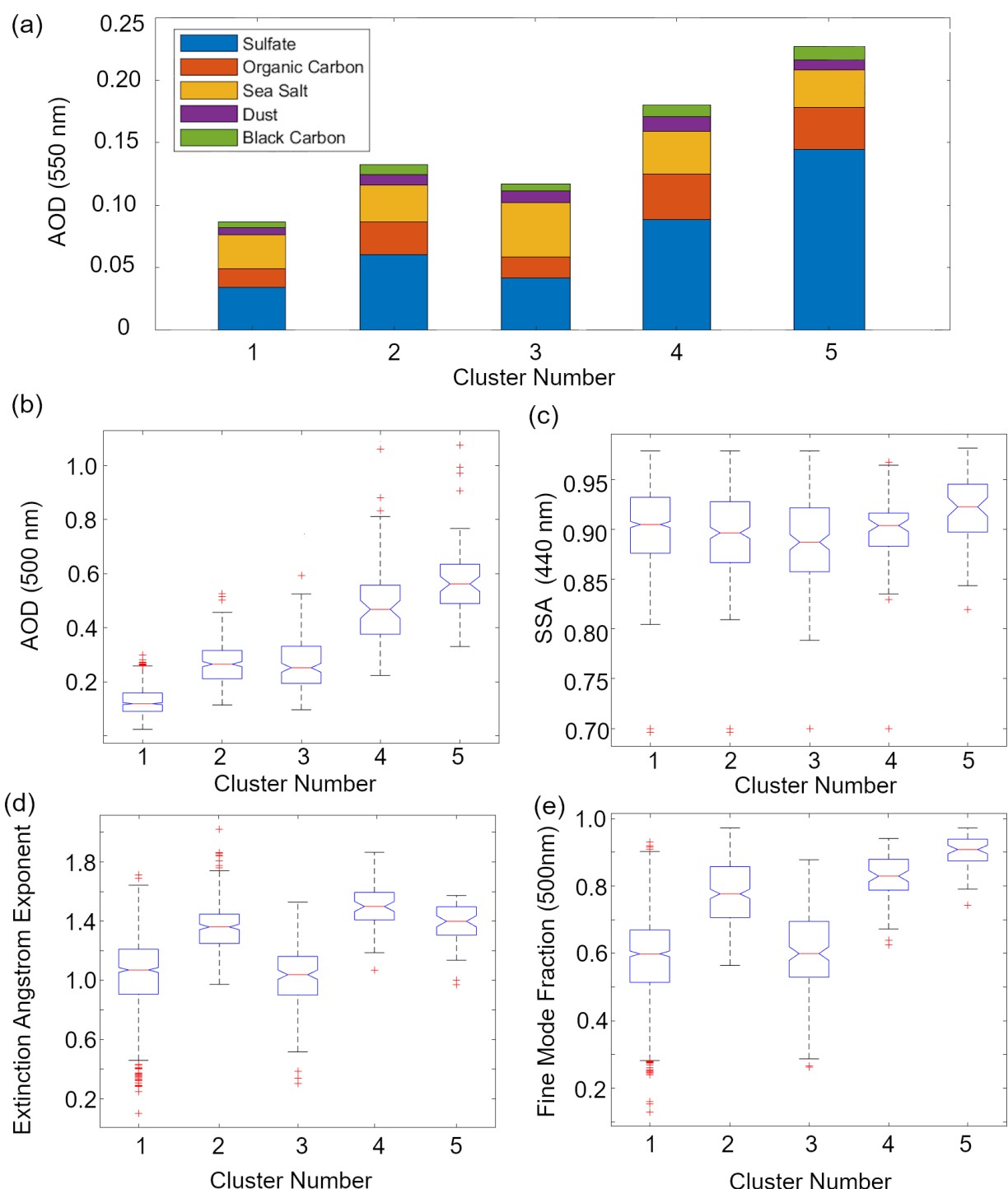

**Figure 7:** (a) Average compositional contributions to aerosol optical depth (AOD at 550 nm)
from MERRA-2 per identified cluster (counts per cluster from 1 to 5: 830, 284, 166, 74, 65).
Boxplots of AERONET (b) total AOD (500 nm), (c) single scattering albedo (SSA at 440 nm),
(d) extinction angstrom exponent (EAE at 440 nm – 870 nm total), and (e) fine mode fraction
(FMF at 500 nm) per cluster.
3.3.2   Air Mass Types
Air masses have been classified in previous studies based on their AOD, EAE, FMF, and SSA
values (e.g., Lee et al., 2010 and Aldhaif et al., 2021). The criteria from different studies (Table
2) were applied per cluster. Median total AOD of cluster 1 (0.12) was less than 0.2 (Fig. 7b),
which is the threshold for sea salt sources. Half of the data points in cluster 1 also fall below the
threshold for clean environments (AOD < 0.1) (Sorooshian et al., 2013). Based on its median
EAE (1.07, where EAE < 1 is coarse and EAE >1 is fine) and FMF (0.60) values (Fig. 7d and
7e), cluster 1 is a mixture of fine and coarse particles. The fine Cluster 1 is the only cluster with a
median that meets that threshold value for clean marine sources (AOD < 0.2), and we know from
Sect. 3.3.1 that its average VSD magnitude was greater for the coarse fraction and that its sea salt
contribution to total AOD was second greatest among the clusters. Thus, most probably, cluster 1
is a background clean marine source, since it also is predominant throughout the seasons (Fig.
6b). This makes sense given the proximity of the ocean to Metro Manila from both the east and
the west. The median SSA (0.90 at 440 nm) for cluster 1 (Fig. 7c), however, suggests the
presence of absorbing particles most probably due to high black carbon in the local source (Cruz
et al., 2019) that is mixed in with this generally clean marine source.
Most of the data from the other clusters all fall in the polluted category (Table 2), based on their
median total AODs (>0.1) (Fig. 7b). Cluster 2 has a median FMF value of 0.78 (Fig. 7e), which
suggests that most of the particles in this air mass are in the fine fraction. They are, however, not
sufficiently dominant in the aerosol for them to be typical of urban/industrial sources. The
average VSDs (Fig. 6a) of cluster 2 similarly suggest that their relative accumulation mode
magnitude is higher than the coarse magnitude, but not much higher. Like cluster 1, cluster 2 is
also more evenly distributed across the seasons (Fig. 6b). The median SSA for cluster 2 (0.90 at
440 nm) is also similar to the SSA of cluster 1 (Fig. 7c) where the local and background particles
are mixed. Cluster 2 could be a fine polluted background source superimposed on the dominant
marine source. Metro Manila is a megacity with continuous and large amounts of sources that
could be, due to its proximity to the ocean, interacting with the background.
Based on its median EAE value (1.04) (Fig. 7d), cluster 3 is mixed but mostly in the coarse
fraction, consistent with its VSD profile (Fig. 6a) which has the highest coarse magnitude (FMF
= 0.60) compared to the other clusters. The contribution of data from September to February is
greatest in cluster 3, consistent with expected coarser particles during this period when the winds
are initially shifting from the southwest before becoming more northeasterly, as previously
noted. Median SSA (0.89 at 440 nm) was lowest for cluster 3 (Fig. 7c), this and the relatively
high coarse particle contribution suggests cluster 3 as a possible dust source based on past
studies (Lee et al., 2010). This air mass can be a mixture of local sources and transported dust air
masses, the large sea salt contribution (~37%) to total AOD (Sect. 3.3.1) can be related to long-
range transport.
Both clusters 4 and 5 have median total FMF (0.83 and 0.91) (Fig. 7e) values exceeding the mark
(> 0.8, Table 2) for urban/industrial air masses. Combining this and results from the previous
sections confirms that cluster 4 can be an urban/industrial source given that it had the highest
median accumulated mode peak and organic carbon contribution (~20%) to total AOD among
the clusters. The median SSA for cluster 4 (0.90 at 440 nm) was similar to the median SSA of
clusters 1 and 2 (Fig. 7c), but the maximum SSA value for this cluster was lowest in general
among all the clusters suggesting cluster 4 has the net most absorptive effect. The cluster 4 air
mass is probably from local sources and transported biomass burning emissions. The high
median EAE (1.40, Fig. 7d) may be associated with aerosol particles due to biomass burning
(Deep et al., 2021).
Cluster 5 had the highest median total AOD (0.56) and FMF (0.91) values (Fig. 7b and 7e). It
also had the highest sulfate contribution (~64%) to total AOD (Fig. 7a), the highest median SSA
(0.92 at 440 nm, thus most reflective particles among the clusters) (Fig. 7c), and a shifted
accumulation mode peak (Fig. 6a). These characteristics suggest that cluster 5 is a possible cloud
processing air mass (Eck et al., 2012). The larger peak in the accumulation mode is possibly the
cloud signature. Previous studies have attributed this larger mode to cloud processing due to the
conversion of $SO_2$ to sulfate (Hoppel et al., 1994). Cloud processing is a major source of sulfate
(Barth et al., 2000).
The distribution of the air masses based on the abundance of the VSD profiles per cluster suggest
prevalent clean marine (58% of the total VSD counts) and background fine polluted (20%) air
masses over Metro Manila. The mixed dust (12%), urban/industrial (5%), and cloud processing
(5%) air masses contribute 22% altogether. We can investigate more deeply and look at specific
case studies that can better describe the air masses identified here.
**3.4    Case Studies**
Selected case studies are used to highlight periods with the highest AOD values and strongest
clear sky (no rain and heavy clouds) daytime aerosol particle sources within the sampling period.
As such, the clusters that are associated with the selected case studies are the clusters (3-5) with
higher VSD concentration magnitudes.
3.4.1    Long Range Transport of Smoke
Both cases of long-range transport of smoke discussed below have similar VSDs (Fig. 8a and 9a)
to the urban/industrial cluster VSD (cluster 4, Fig. 6a). Organic carbon was the dominant
contributor to AOD (Fig. 8b and 9b) for both long-range transport cases. The first of two events
occurred around 1 April 2020 with smoke presumed to come from East Asia. The VSD of this
specific case (Fig. 8a) is most like the urban/industrial cluster (cluster 4 in 3.3.2, Fig. 6a) because
of the high magnitude of its accumulated mode peak, its timing (April), and the enhanced
organic carbon contribution to AOD in the area (Fig. 8b). Though the absolute black carbon
contribution to AOD was highest here compared to the other case studies, and in general for the
AERONET data, it was organic carbon that was more prevalent in terms of contribution to total
AOD. Smoke is comprised of both soot carbon and organic carbon, amongst other constituents
(Reid et al., 2005).

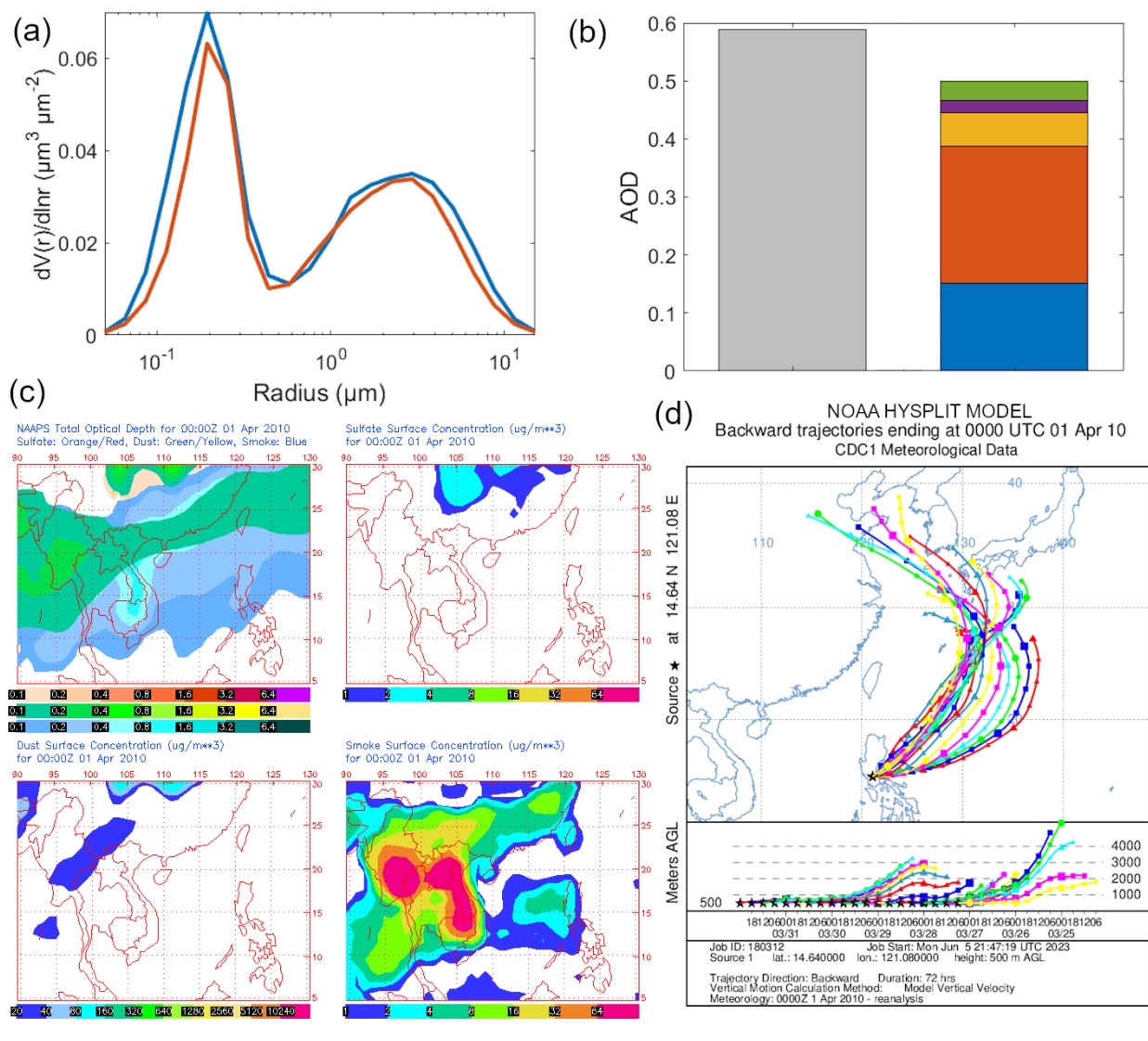


Apr 1 02:30:45 2010    NRL/Monterey Aerosol Modeling

**Figure 8:** Case study of long-range transport (smoke – East Asia) around 1 April 2010. (a)
AERONET VSDs at (blue) 00:01 and (red) 00:26 UTC, (b) AOD from AERONET (gray:
median AOD  at 500 nm) and MERRA-2 hourly (green: black carbon, violet: dust, yellow: sea
salt, orange: organic carbon, blue: sulfate) compositional contributions to AOD (550 nm) closest
in time to 00:01 UTC, (c) NAAPS maps of total and compositional hourly AOD (orange/red:
sulfate, green/yellow: dust, blue: smoke) and  sulfate, dust, and smoke surface concentrations at
00:00 UTC, and (d) HYSPLIT seven-day back-trajectories arriving at Manila Observatory at
00:00 UTC.
The smoke contribution to AOD from NAAPS (Fig. 8c) for the first smoke case was visible in
the Philippines (0.2) and seemed to come from East Asia were the smoke contribution to AOD
was greater (reaching 0.8) especially in Peninsular Southeast Asia. Smoke surface concentrations
were also widespread (Fig. 8c) with greatest concentrations in East Asia that reached the
Western Philippines, though seemingly disconnected over the sea. There were observed biomass
burning emissions in the Peninsular Southeast Asia (southern China, Burma, and Thailand) at
this time (Shen et al., 2014). The direction of the air mass coming into Metro Manila was from
the northeast, which curved from the west in the direction of East Asia based on HYSPLIT back-
trajectories (Fig. 8d).
The second smoke case was on 15 September 2009 with the source being Southeast Asia. The
back-trajectories of this case study (Fig. 9d) are from the southwest of the Philippines, and in the
direction of the Malaysia and Indonesia. NAAPS maps likewise show elevated AOD,
specifically smoke contribution to AOD (Fig. 9c), as well as enhanced smoke surface
contributions in the area around Metro Manila for this second smoke case study. The observed
AOD and smoke surface concentration increased specifically from the southwest of the
Philippines in the same direction of the back-trajectories. There were fires in the lowland (peat)
forests of Borneo around this time (NASA, 2009). MERRA-2 AOD contributions for this case
were greatest due to organic carbon as well as sulfate (Fig. 9b), and the absolute black carbon
contributions were greatest compared to other cases. The VSD of this smoke case from Southeast
Asia (Fig. 9a) resembled that from long-range transported smoke from East Asia (Fig. 8a) and
the urban/industrial air mass (cluster 4, Fig. 6a). This case occurred in the afternoon, which was
the prevalent time that the urban/industrial air mass was observed (Fig. 6c).

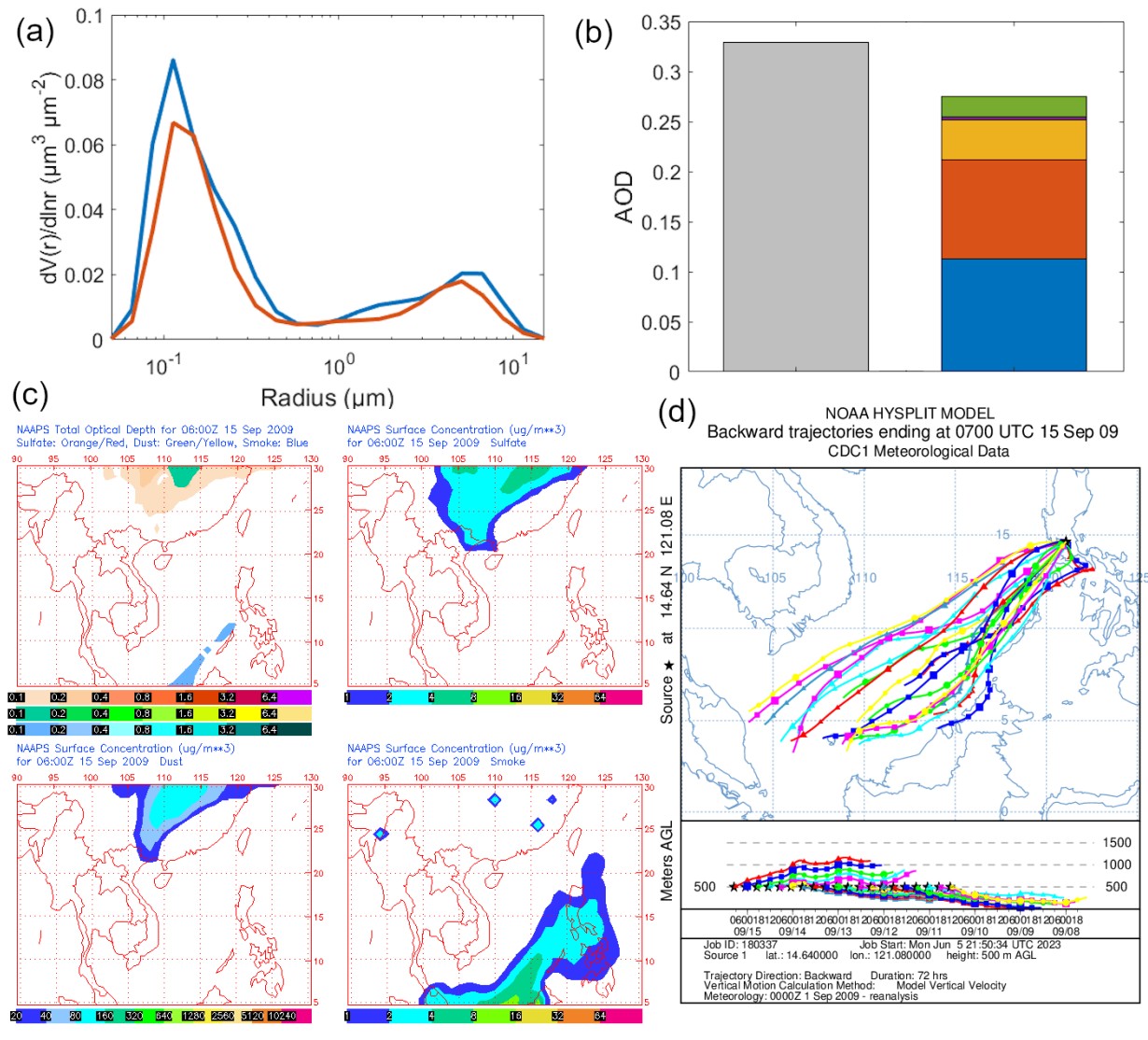

**Figure 9:** Case study of long-range transport (smoke – Southeast Asia) around 15 September 2009. (a) AERONET VSDs at (blue) 07:27 and (red) 07:52 UTC, (b) AOD from AERONET (gray: median AOD  at 500 nm) and MERRA-2 hourly (green: black carbon, violet: dust, yellow: sea salt, orange: organic carbon, blue: sulfate) compositional contributions to AOD (550 nm) closest in time to 07:27 UTC, (c) NAAPS maps of total and compositional hourly AOD (orange/red: sulfate, green/yellow: dust, blue: smoke) and sulfate, dust, and smoke surface concentrations at 06:00 UTC, and (d) HYSPLIT seven-day back-trajectories arriving at Manila Observatory at 07:00 UTC.

### 3.4.2   Long Range Transport of Dust

The VSD of this specific case on 24 March 2018 (Fig. 10a) was most similar to the mixed dust cluster (cluster 3), which had a mixed size distribution but a more dominant coarse contribution. This is consistent with the most dominant contribution to AOD in the area, which was sea salt and dust (Fig. 10b). The back-trajectories were from East Asia around the same latitude as Taiwan (Fig. 10d). That area, at that time, had increased AOD in general from sulfate and dust

(Fig. 10c). The AOD from both AERONET and MERRA-2 (Fig. 10b) are lower than 0.3 (the
AOD threshold for dust in other studies, Table 2) because of the long distance from the source
(thousands of kilometers). The dust and sulfate seemed to have been transported to Metro Manila
from East Asia based on the NAAPS sulfate and dust surface concentrations (Fig. 10c).

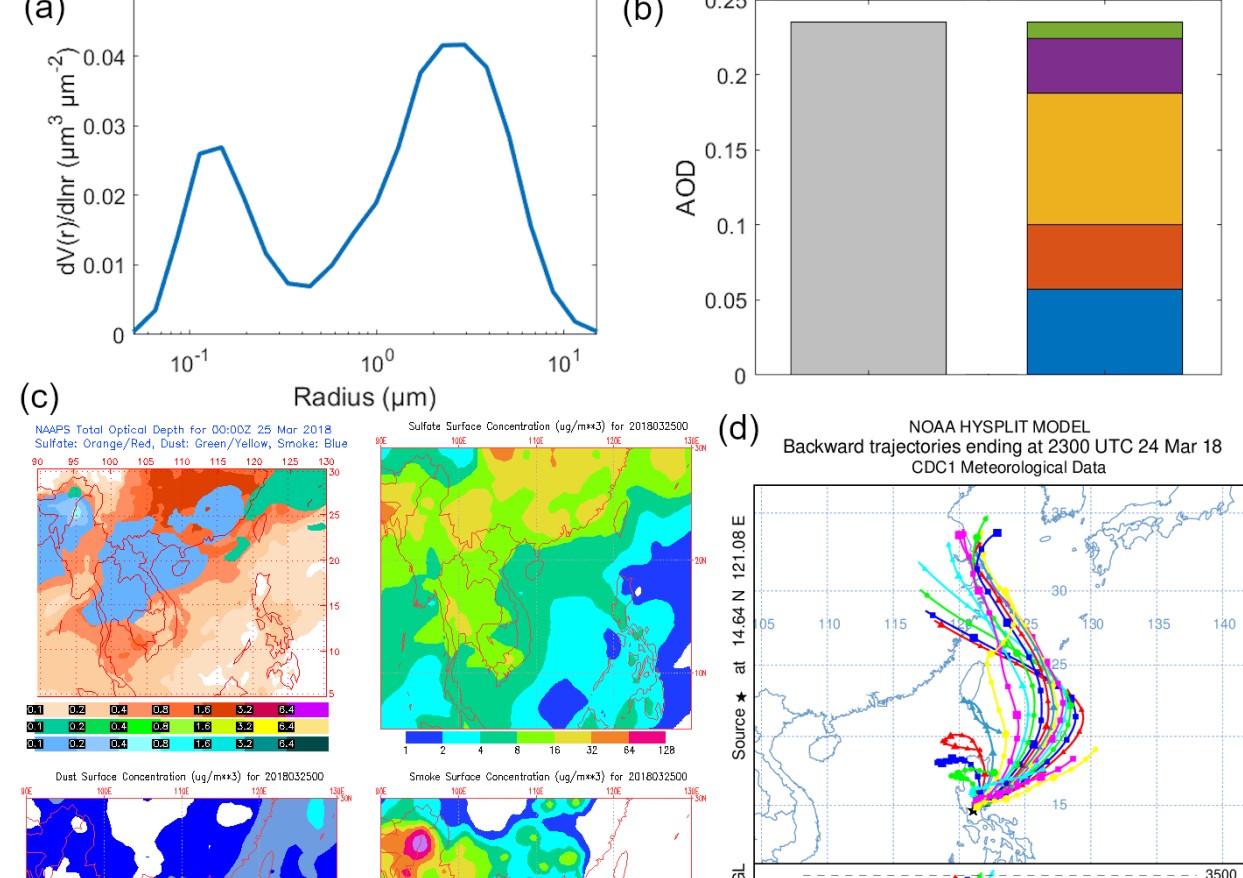

**Figure 10:** Case study of long-range transport (dust) around 24-25 March 2018. (a) AERONET
VSD at (blue) 23:23 UTC, (b) AOD from AERONET (gray: AOD  at 500 nm) and MERRA-2
hourly (green: black carbon, violet: dust, yellow: sea salt, orange: organic carbon, blue: sulfate)
compositional contributions to AOD (550 nm) closest in time to 23:23 UTC, (c) NAAPS maps of
total and compositional hourly AOD (orange/red: sulfate, green/yellow: dust, blue: smoke) and
sulfate, dust, and smoke surface concentrations at 00:00 UTC on March 25, and (d) HYSPLIT
seven-day back-trajectories arriving at Manila Observatory at 23:00 UTC.

3.4.3    Cloud Processing
Sulfate dominated the AOD (Fig. 11b) for this case on 26 August 2009 in the area around Metro
Manila. This along with its VSD exhibiting a second peak (Fig. 11a) in the accumulation mode
make it very similar to the cloud processing cluster (cluster 5). Sulfate has been known to be
enhanced through chemical productions in clouds and is used as a signature for cloud processing
(Barth et al., 2000; Ervens et al., 2018). Aqueous production of sulfate is significant in areas with
sources and clouds (Barth et al., 2000), and this case study has both. Aside from the high sulfate
contribution to AOD, the cloud fraction (Aqua/MODIS, Terra/MODIS, Fig. S3) is very high
(~100%) in the area of the back-trajectories (Fig. 11d).  Interestingly, there is no regional AOD
elevation observed in the NAAPS maps (Fig. 11c) for this time. There are increased surface
smoke and sulfate levels in East Asia as well as southwest of the Philippines, and though the
back-trajectories do show a northeastward direction, they do not reach far enough into mainland
East Asia. It is possible that even while there are known regional sources of sulfate in Southeast
Asia (Smith et al., 2011; Li et al., 2017), this case could be local to the Philippines. There is in
fact a large power plant northwest of Metro Manila (Jamora et al., 2020).

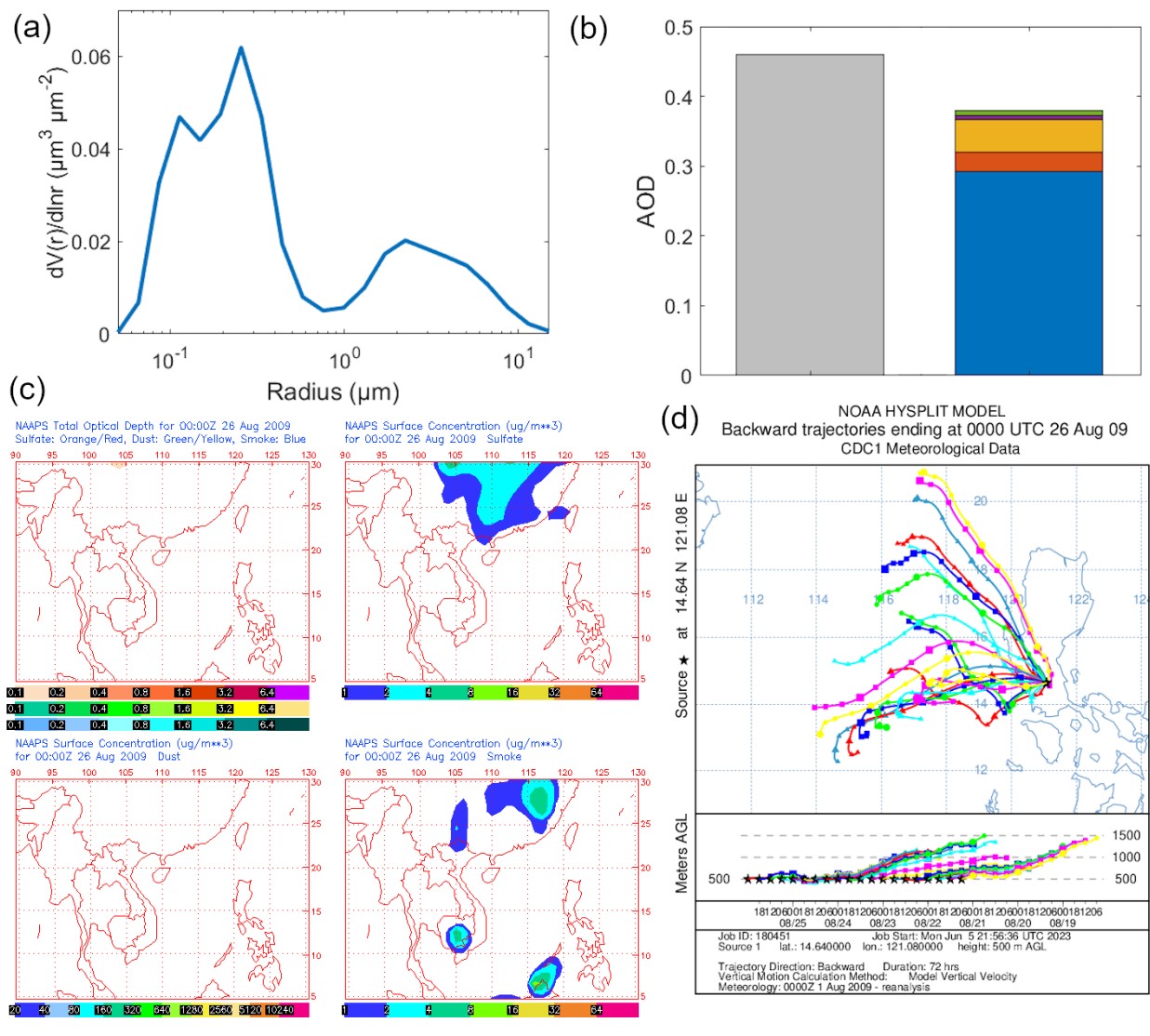

**Figure 11:** Case study of cloud processing on 26 August 2009. (a) AERONET VSDs at 00:18 UTC, (b) AOD from AERONET (gray: median AOD at 500 nm) and MERRA-2 hourly (green: black carbon, violet: dust, yellow: sea salt, orange: organic carbon, blue: sulfate) compositional contributions to AOD (550 nm) closest in time to 00:18 UTC, (c) NAAPS maps of total and compositional hourly AOD and contributions and smoke surface concentrations at 00:00 UTC, and (d) HYSPLIT seven-day back-trajectories arriving at Manila Observatory at 00:00 UTC.

## 3.5    EOF Analysis of AOD in Southeast Asia

The air masses in Metro Manila are influenced by regional sources which were identified through EOF analysis of AOD. Three principal components (PC, Fig. 12) explained most of the data variance (73.77%) (Fig. 12a) and were all well-separated from each other and are therefore most probably the major distinct aerosol particle sources in the region.  They will be the focus of the subsequent discussion.

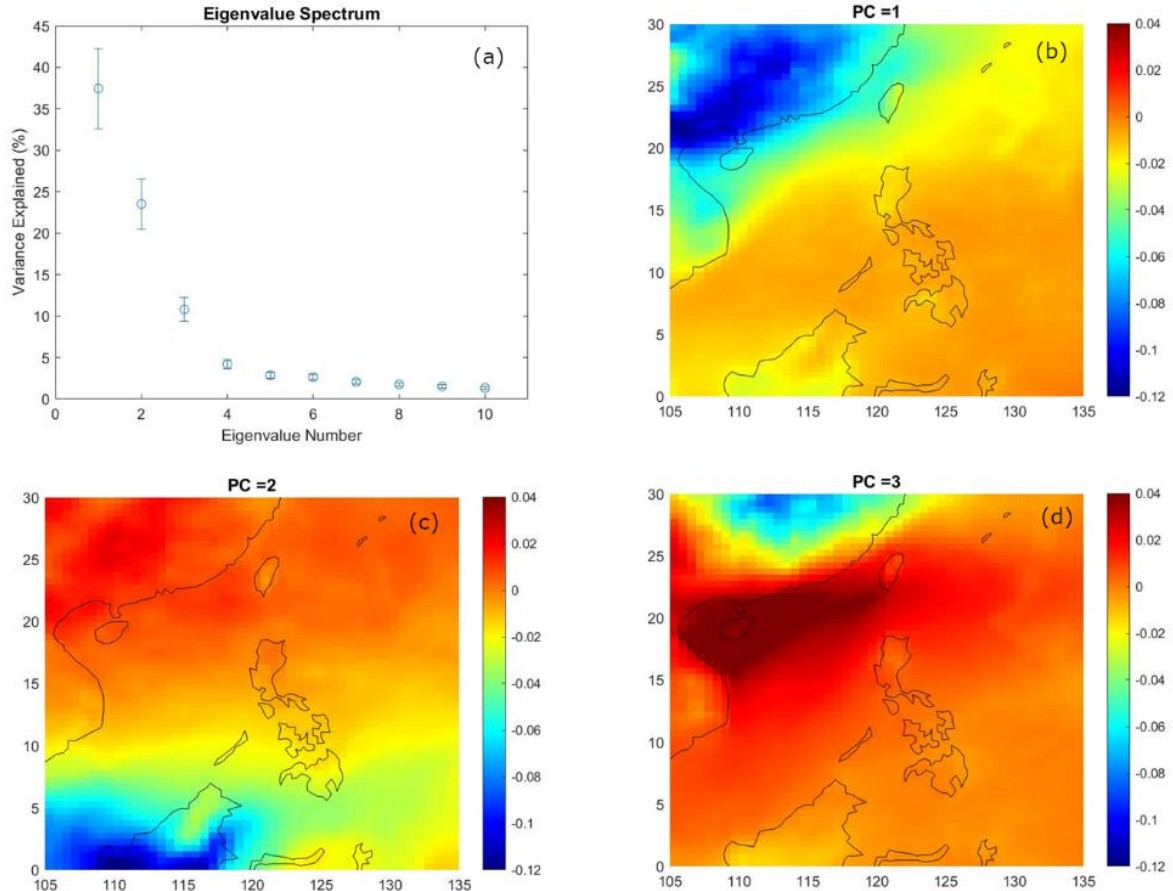

**Figure 12:** Results of the singular value decomposition. (a) Eigenvalue spectrum of the first ten
eigenvalues, (b-d) maps of the coefficients of regression AOD anomalies onto the first three
principal components.
The first PC explains 37.46% of the data variance (Fig. 12a) and, based on the map of the
regression coefficients (Fig. 12b), separates mainland East Asia from the Philippines and
Indonesia. East Asia is a globally recognized source for high AOD (Li et al., 2013), and its
contribution to particles in Southeast Asia possibly corresponds to the first PC. The second PC
explains 25.51% of the data variance (Fig. 12a) and separates southern Southeast Asia from
northern Southeast Asia at around 15°N (Fig. 12c). Southern Southeast Asia is a known regional
source of aerosol particles due to biomass burning (Cohen et al., 2017) and could be associated
with the second PC. The third PC explains 10.80% of the data variance (Fig. 12a) and separates
northern East Asia from southern East Asia mainland and the rest of Southeast Asia (Fig. 12d).
To gain confidence in the association of the PCs with their sources, we present correlation maps
between the first three PCs to the fractional contributions of sulfate and organic carbon to AOD
for the entire dataset.
The correlation maps of the first PC and the sulfate contribution to AOD (Fig. 13a and 13d)
show high and statistically significant correlations (gray areas) in mainland East Asia and
Taiwan, parts of western Philippines and Borneo, which are the probable sulfate sources. Clues
from the mean monthly wind vector maps in April (Fig. 14a and 14d) and mean monthly AOD in

either March or April (Fig. S3c or S3e) most resembling the features of regression map of the first PC (Fig. 12b) and the PC time series peaking in March (Fig. S4) together suggest that the first PC may be associated with air masses that are present around March or April. Emissions sources and meteorology that are dominant during the peak dates in the PC time series offer clues to the attribution of each PC. The Southeast Asia region and the Philippines is influenced by the monsoon systems (Coronas, 1920; Matsumoto et al., 2020) and February to March is the time when the winds are transitioning from the northeasterly to easterly. The first PC could be affected by the easterly winds, which are dominant around March when its PC values peaked. The higher-level winds (free troposphere) (Fig. 14a) in April are from the west in mainland East Asia and are from the east in the Philippines and it is possible that the different wind regimes are distinguishing the sulfate sources in East Asia and the Philippines and beyond. Sulfate is a known product of industry in East Asia (Smith et al., 2011; Li et al., 2017) while the West Luzon and West Visayas islands have large power plants (Jamora et al., 2020).

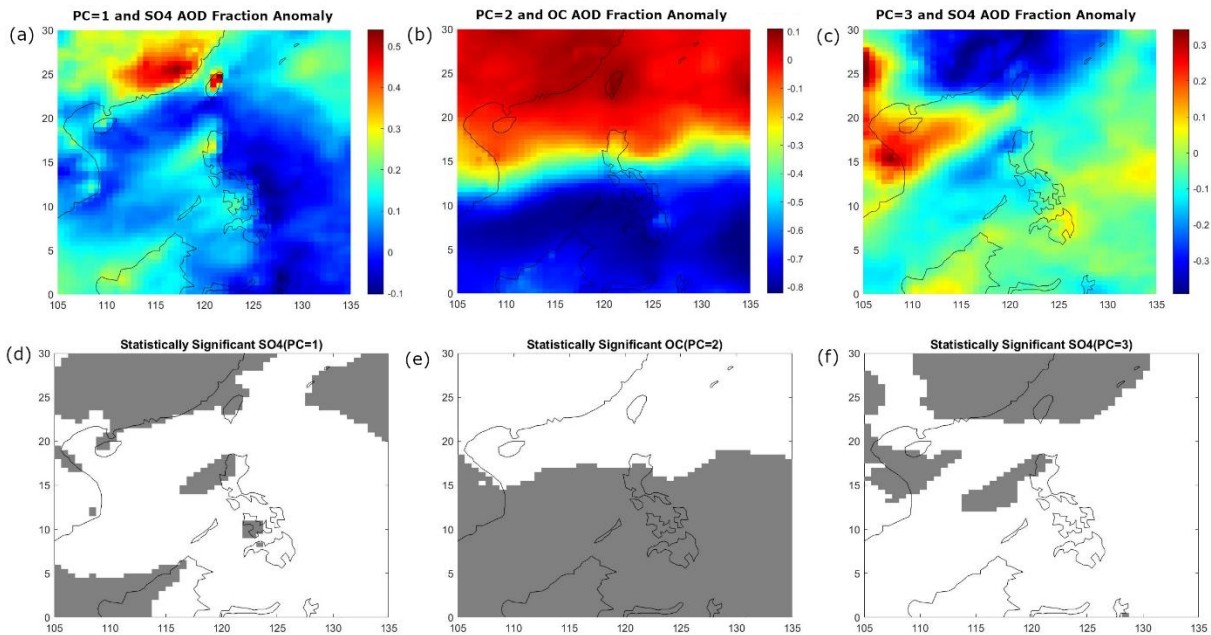

**Figure 13:** Correlation coefficients of principal components with (a/c) sulfate AOD fraction and (b) organic carbon AOD fraction. Statistically significant (90%, d-f) areas are shaded gray.

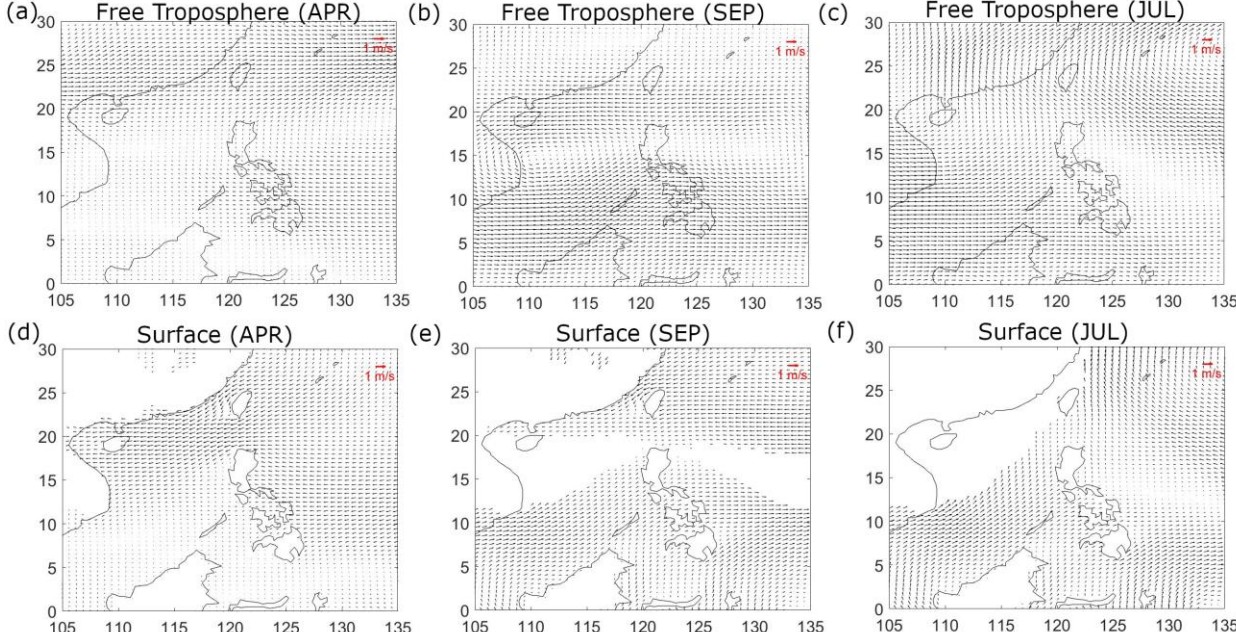

**Figure 14:** Monthly averaged winds for (a & d) April, (b & e) September, and (c & f) July from
MERRA-2 at (725 hPa, a-c) the free troposphere approximate and at (1000 hPa, d-f) the surface.
The correlation maps of the second PC and the OC contribution to AOD (Fig. 13b and 13e) show
high and statistically significant correlations from 0°N to 15°N. The large magnitude of the
correlation coefficient (gray areas in Fig. 13b) stands out in southern Southeast Asia and is the
potential OC source. In this case, it is known that Indonesia is a major source of biomass burning
during its fire season (Glover and Jessup, 1998), and thus the local significance established in the
southern Southeast Asia is most likely due to the Indonesia biomass burning source. The burning
season in Indonesia is from August to October, and that is the same time when the AOD values
peak in the area (Fig. S3h, S3i, and S3j), as well as the peak of the second PC in the time series
(Fig. S4). Winds are usually from the southwest and west due to the southwest monsoon from
September to October, when the second PC peaked, and thus the second PC may be related to the
southwest monsoon. During the same time the surface and free troposphere mean monthly winds
(Fig. 14b and 14e) are from the southwest (in the general direction of Indonesia) towards the
south portion of Southeast Asia and thus corroborate the observation that the second PC may be
highlighting the regional effect of the Indonesia forest fires. Of interest is the line of separation
of the northern and southern Southeast Asia in the principal component that is within the area of
the monsoon trough (Wang et al., 2007).  This line is also evident in the surface and the free
troposphere maps where the southwest winds from the area of Indonesia meet the easterlies in
north Southeast Asia (Fig. 14b and 14e) and which thus appears to be limiting the dispersion of
the biomass burning emissions to southern Southeast Asia.
The third PC was also well correlated to the sulfate AOD fraction though, compared to the first
PC correlation maps, there were distinctions between the northern and southern East Asia
regions (Fig. 13c and 13f). The local Philippine source still came out in the correlation maps as a
significant source.  It was not clear from the PC time series (Fig. S4), which showed peaks in the
third PC in February, how the dates were related to the PC profile. The free troposphere winds in
July (Fig. 14c), as well as the AOD monthly mean map in July (Fig. 14c), however, showed
more similarities to the third PC regression map. Both showed a delineation between the
northern East Asia and southern East Asia (including Hong Kong) features. Mean winds (Fig.
14c) in the free troposphere are from the west, due to the southwest monsoon, in the area around
the Philippines, and they were from the northeast in north Southeast Asia.  The interface of the
winds is within the approximate location of the monsoon trough in July (Wang et al., 2007), and
it is thus possible that the monsoon trough is causing the separation of the sulfate sources. This
could be investigated further. The monsoon trough has been noted to scavenge aerosol particles
from southern Southeast Asia (Reid et al., 2013). It is evident from the analysis that meteorology
affects the transport and processing of aerosol particles in region which along with local sources
contribute to the aerosol composition in Southeast Asia (Cruz et al., 2019; AzadiAghdam et al.,
2019; Braun et al., 2020; Hilario et al., 2020b; Hilario et al., 2022).
**4.   Conclusion**
Metro Manila has both urban and industrial local sources known to contribute to the dominance
of fine mode particles in its air (Cruz et al., 2019). Ten years of AERONET data in Manila
Observatory suggest that aerosol particles in Metro Manila were mixed in size but with a
prevalent fine mode fraction (>50% FMF) throughout the year. Background clean marine aerosol
particles (58% of the time) and fine polluted aerosol particles (20% of the time) were the most
dominant clear sky day sources impacting the atmospheric column over Metro Manila based on
cluster analysis of volume size distributions. The proximity of Metro Manila to the sea, both in
the east and west, along with local sources, transportation being the most prominent, together
contribute to the prevalence of the marine and fine particles. The prevalence of marine particles
could explain the relatively small AOD values in Metro Manila compared to other Southeast
Asian megacities (Reid et al., 2013).
Regional sources and meteorology also impact monthly aerosol optical depth trends in Metro
Manila from EOF analysis. Biomass burning from Borneo and Sumatra emerged in the study as
the second most prevalent regional anthropogenic aerosol particle source in Southeast Asia.
Though the monsoon trough limits the dispersion of aerosol particles throughout the entire
Southeast Asia, biomass burning emissions impact southern Southeast Asia including Metro
Manila during the southwest monsoon (July to September). The monsoon winds facilitate the
transport of fine particles during the peak burning season in Borneo and Sumatra (August-
September). This is experienced in Metro Manila as higher than usual aerosol particle loadings
around the same period (August to October). Climatologically, August was also when there were
particles with the greatest fine mode fractions that were relatively absorbing and non-
hygroscopic possibly due to increased organic and elemental carbon fractional contributions.
Though not as strong a source as the Borneo and Sumatra case, the peninsular Southeast Asia
burning season (March-April) also contributed to extreme aerosol particle concentrations over
Metro Manila.
High aerosol particle loadings due to transported dust, probably from East Asia, were observed
in Metro Manila during the transition period between the southwest and northeast monsoons and
during the northeast monsoon (December to February). These extreme events are transient
because the lowest median aerosol particle loadings of the year were observed during the
northeast monsoon when annual wind speeds were highest. Particles then were observed to be
largest in diameter, with the greatest coarse fraction contribution, relatively high absorptivity,
and most hygrosocopicity, compared to other months of the year. This is probably due to
constituents other than soot, especially aged dust (Kim and Park, 2012; Geng et al., 2014) and
sea salt which the northeast winds appear to be bringing in from the general direction of the
Luzon Island and the Philippine Sea (West Pacific Ocean).
Cloud processing is one of the cases that were linked to very high aerosol particle loading in
Metro Manila. This is associated with sulfate sources, which appear more localized in nature
because of a power plant nearby. This sulfate source seems to be distinct from the industrial
sulfate air mass from East Asia, which is the most dominant regional aerosol particle source in
Southeast Asia (Li et al., 2013). Winds appear to limit the mixing of this notable East Asia air
mass with local industrial sources in the region including the Philippines and Indonesia.
The formation of cloud systems in Southeast Asia is complex due to intersecting large- and
small-scale mechanisms. Additionally, the interaction of particles and clouds in Southeast Asia is
not yet well understood. In Metro Manila, both topography and meteorology affect aerosol
particle distribution (Cruz et al., 2023). This baseline study on the aerosol particle characteristics
in Metro Manila and in regional Southeast Asia shows how meteorology impacts varied aerosol
particle sources (e.g., sulfate, elemental carbon, and organic carbon) and their distribution in the
region. This can help in mitigating aerosol particle sources in the region and in the deepening of
the understanding of the relationship of aerosol particles, meteorology, and clouds.

**Data availability**
Aerosol Robotic Network (AERONET) (2020), Version 3 Direct Sun Algorithm, Site: Manila
Observatory, Philippines, Accessed: **[*28 September 2020*]**, https://aeronet.gsfc.nasa.gov/cgi-
bin/webtool_aod_v3?stage=3®ion=Asia&state=Philippines&site=Manila_Observatory&plac
e_code=10&if_polarized=0
Aerosol Robotic Network (AERONET) (2020), Version 3 Direct Sun and Inversion Algorithm,
Site: Manila Observatory, Philippines, Accessed: **[*28 September 2020*]**,
https://aeronet.gsfc.nasa.gov/cgi-
bin/webtool_inv_v3?stage=3®ion=Asia&state=Philippines&site=Manila_Observatory&place
_code=10&if_polarized=0
Multi-angle Imaging SpectroRadiometer (MISR) Jet Propulsion Laboratory (2018), Level 3
Component Global Aerosol product in netCDF format covering a month V004, Accessed: **[*22
November 2021*],** https://search.earthdata.nasa.gov/
Global Modeling and Assimilation Office (GMAO) (2015), MERRA-2 inst3_3d_asm_Np: 3d,3-
Hourly, Instantaneous, Pressure-Level, Assimilation, Assimilated Meteorological Fields V5.12.4,
Greenbelt, MD, USA, Goddard Earth Sciences Data and Information Services Center (GES
DISC), Accessed: **[*10 March 2021*]**, https://doi.org/10.5067/QBZ6MG944HW0
Global Modeling and Assimilation Office (GMAO) (2015), MERRA-2 tavg1_2d_flx_Nx: 2d,1-
Hourly, Time-Averaged, Single-Level, Assimilation, Surface Flux Diagnostics V5.12.4,
Greenbelt, MD, USA, Goddard Earth Sciences Data and Information Services Center (GES
DISC), Accessed: **[*10 March 2021*]**, https://doi.org/10.5067/7MCPBJ41Y0K6
Global Modeling and Assimilation Office (GMAO) (2015), MERRA-2 tavg1_2d_csp_Nx: 2d,1-
Hourly, Time-averaged, Single-Level, Assimilation, COSP Satellite Simulator V5.12.4,
Greenbelt, MD, USA, Goddard Earth Sciences Data and Information Services Center (GES
DISC), Accessed: **[*13 July 2021*]**, https://doi.org/10.5067/H0VVAD8F6MX5
Nguyen, P., E.J. Shearer, H. Tran, M. Ombadi, N. Hayatbini, T. Palacios, P. Huynh, G.
Updegraff, K. Hsu, B. Kuligowski, W.S. Logan, and S. Sorooshian, The CHRS Data Portal, an
easily accessible public repository for PERSIANN global satellite precipitation data, Nature
Scientific Data, Vol. 6, Article 180296, 2019, Accessed: **[*11 March 2021*],**
https://doi.org/10.1038/sdata.2018.296

**Author contributions**

GRL and AS designed the experiment. NL, SNU, GRL, GFG, HJO, JBS, and MTC, carried out
various aspects of the data collection. GRL, AS, JBS, MOC, MRH, CC, and LDG conducted
analysis and interpretation of the data. GRL prepared the manuscript draft with contributions
from the coauthors. AFA, LDG, MRH, GRL, and AS reviewed and edited the manuscript. AS
led the management and funding acquisition. All authors approved the final version of the
manuscript.

**Competing interests**

We declare that Armin Sorooshian is a member of the editorial board of Atmospheric Chemistry
and Physics. The peer-review process was guided by an independent editor, and the authors have
also no other competing interests to declare.

**Acknowledgements**

The authors acknowledge support from NASA grant 80NSSC18K0148 in support of the NASA
CAMP[2]Ex project, in addition to ONR grant N00014-21-1-2115. We acknowledge the US Naval
Research Laboratory for providing the AERONET instrument. We acknowledge the use of
imagery from the NASA Worldview application (https://worldview.earthdata.nasa.gov), part of
the NASA Earth Observing System Data and Information System (EOSDIS).

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
