# Peer review of "An Emerging Aerosol Climatology via Remote Sensing over Metro Manila, Philippines"

_EGUsphere, 2023_

## Author Response (AR1)

Response: We thank the two reviewers for the thoughtful suggestions and constructive criticism that have helped us to continue to improve our manuscript. Below we provide responses to referee comments and suggestions in blue font. All changes to the manuscript can be identified in the latest version submitted as well as in the marked-up manuscript at the end of this document.

Referee Comment: "An Aerosol Climatology via Remote Sensing over Metro Manila, Philippines"
Anonymous Referee #1
April 25, 2023

General Comments:

"Climatology" is generally referred to data over at least 30 years. In this study, the Authors present far fewer years, and these data are therefore considered long-term averages. The title needs to be changed. For example, it could be changed to "An Emerging Aerosol Climatology…" as inspired by Holben et al 2001. The Authors present a comprehensive analysis using several different aerosol and meteorological data sets to determine aerosol source partitioning over the Manila metropolitan region. The manuscript uses results from five clustered AERONET aerosol volume size distributions tied to MERRA-2 speciated AOD to determine the type of aerosol particles over the Manila region. At times, it is not clear how MERRA-2 data are being matched to AERONET data. As expected, Manila is dominated by coarse mode sea salt with fine mode pollution and sometimes cloud processed aerosols. Manila is impacted episodically by other aerosol sources such as smoke and pollution from other transboundary sources.

Overall, the manuscript is written and organized well; however, the data use and explanations resulted in several questions which are described below and may result in modifications to the data analysis. The questions below need to be clarified and changes supported by citations before publication.

Response: We thank the reviewer for this summarized perspective. As detailed below, we took actions to remedy some of these concerns. For instance, we did revise the title to now be "An Emerging Aerosol Climatology via Remote Sensing over Metro Manila, Philippines"

Detailed Comments:

Lines 24 and 25: Please combine these two sentences for a more complete statement. For example, "Aerosol particles…are challenging to characterize" due to "the diverse…"

Response: The two sentences have been combined: "Aerosol particles in Southeast Asia are challenging to characterize due to their complex life cycle within the diverse topography and weather in the region."

Line 26: Is it a climatology by definition? It seems more appropriate to name it a "long-term average" or perhaps "emerging climatology" as coined by Holben et al., 2001.

Response: The technical definition of climatology involves thirty years of data. The lack of long-term aerosol data does not make this possible yet for Metro Manila. However, some authors have used the word climatology to describe analysis of data from 10 (Gui et al., 2022; Banerjee et al., 2021) to 20 years (Kinne, 2019). Others have referred to ten-year data analysis as "decadal climatology" (Tandule et al., 2020). To avoid confusion about the terminology, the word "emerging" was inserted before climatology.

Lines 27 to 30: These sentences should be combined and revised to produce a more complete statement.

Response: The sentences have been combined, and the timeframe of the high AOD values now include the months until October: "Aerosol optical depth (AOD) values were highest from August to October, partly from fine urban aerosol particles, including soot, coinciding with the burning season in Insular Southeast Asia when smoke is often transported to Metro Manila during the southwest monsoon."

Lines 30-37: MERRA-2 aerosol particle composition data are on the monthly basis (as discussed in Section 2.1.2) rather than determined for the nearest time to the AERONET size distribution retrieval in which the size distributions could change significantly in aerosol urban and transported plumes. What is the difference between "fine polluted," "mixed polluted," "urban/industrial," and "cloud processed" since they are all in the fine mode.

Response: Hourly AOD data (M2T1NXAER) were used for comparison to the nearest AERONET volume size distribution data. The information in Section 2.1.2 (end of paragraph 1) was updated to include this dataset information: "; and M2T1NXAER Aerosol Diagnostics (1-hourly from 00:30 UTC time-averaged) for Total AOD and speciated AOD (Sulfate, Black Carbon (BC), Organic Carbon (OC), Dust, and Sea Salt)." Table 1 was likewise updated.

Fine polluted is the more general terminology. The first four mentioned air mass types in Table 2 are the most general (based on size fraction and clean or polluted conditions). More specific classifications based on aerosol particle sources are included after those four general ones. The urban/industrial air mass type here refers to local combustion along with long-range transported biomass burning. The terminology mixed/polluted has been dropped and replaced by a potential dust source (mixed dust). The cloud processed air mass is a more specific type of fine polluted air mass which has an accumulation mode peak that's slightly larger. This information has been added to the abstract, as shown here: "The following are the other particle sources over Metro Manila: fine polluted (20%), mixed dust (12%), urban/industrial (5%), and cloud processing (5%)."

Lines 114-116: Schuster reference is not very appropriate here. More appropriate is Dubovik and King, 2000, Dubovik et al. 2000, Dubovik et al. 2002, and Dubovik et al., 2006.

Response: The following references were used instead: (Dubovik and King, 2000; Dubovik et al., 2000; Dubovik et al., 2002; Dubovik et al., 2006).

Lines 132-151: Why not use the hourly MERRA-2 product (M2T1NXAER) to match up with AERONET?

Response: Yes, the hourly product (M2T1NXAER) was actually used, and it was just not indicated in the methodology previously. This information now has been included in this section: "; and M2T1NXAER Aerosol Diagnostics (1-hourly from 00:30 UTC time-averaged) for Total AOD and speciated AOD (Sulfate, Black Carbon (BC), Organic Carbon (OC), Dust, and Sea Salt)." Table 1 was also updated as shown below.

| Parameter | Data Source | Spatial Coverage | Time Coverage |
|---|---|---|---|
| Aerosol Optical Depth (500 nm) | AERONET | 14.635°N, 121.078°E | Jan 2009 - Oct 2018 |
| Asymmetry Factor (440 nm - 1020 nm) | AERONET | 14.635°N, 121.078°E | Jan 2009 - Oct 2018 |
| Extinction Angstrom Exponent (440 nm -870 nm) | AERONET | 14.635°N, 121.078°E | Jan 2009 - Oct 2018 |
| Fine Mode Fraction | AERONET | 14.635°N, 121.078°E | Jan 2009 - Oct 2018 |
| Precipitable Water | AERONET | 14.635°N, 121.078°E | Jan 2009 - Oct 2018 |
| Single Scattering Albedo (440 nm - 1020 nm) | AERONET | 14.635°N, 121.078°E | Jan 2009 - Oct 2018 |
| Refractive Index (Real and Imaginary; 440 nm - 1020 nm) | AERONET | 14.635°N, 121.078°E | Jan 2009 - Oct 2018 |
| Volume Size Distribution | AERONET | 14.635°N, 121.078°E | Jan 2009 - Oct 2018 |
| Low Cloud Fraction (MODIS) | MERRA-2 | 14.3°N - 14.8°N, 120.75°E - 121.25°E | Jan 2009 - Dec 2018 |
| Planetary Boundary Layer Height | MERRA-2 | 14.3°N - 14.8°N, 120.75°E - 121.25°E | Jan 2009 - Dec 2018 |
| Relative Humidity (975 mb) | MERRA-2 | 14.3°N - 14.8°N, 120.75°E - 121.25°E | Jan 2009 - Dec 2018 |
| Sea Level Pressure | MERRA-2 | 14.3°N - 14.8°N, 120.75°E - 121.25°E | Jan 2009 - Dec 2018 |
| Temperature (975 mb) | MERRA-2 | 14.3°N - 14.8°N, 120.75°E - 121.25°E | Jan 2009 - Dec 2018 |
| Wind (975 mb) | MERRA-2 | 14.3°N - 14.8°N, 120.75°E - 121.25°E | Jan 2009 - Dec 2018 |
| Total Extinction Aerosol Optical Depth (550 nm) | MERRA-2 | 14.3°N - 14.8°N, 120.75°E - 121.25°E | Jan 2009 - Dec 2018 |
| Sulfate, Black Carbon, Organic Carbon, Dust, and Sea Salt Extinction Aerosol Optical Depth (550 nm) | MERRA-2 | 14.3°N - 14.8°N, 120.75°E - 121.25°E | Jan 2009 - Dec 2018 |
| Precipitation | PERSIANN | 14.3°N - 14.8°N, 120.75°E - 121.25°E | Jan 2009 - Dec 2018 |

Line 152-161:  It is important to state here that MISR retrievals are much fewer than other LEO and GEO sensors.  How well do the monthly AOD averages from MISR represent the conditions over the Philippines in such a meteorological diverse environment?

Response: The large fraction of cirrus clouds overlapping with lower clouds, dominantly shallow cumuli, is significant in the CAMP[2]Ex domain (Hong and Di Girolamo, 2020). This significantly impacts the number of Level 2 aerosol retrievals, not only for MISR, but all passive instruments, but more so for MISR because of its narrower swath. The sampling error is of course reduced by first computing monthly mean for individual 0.5° grid cells over 10 years of MISR data, followed by taking the mean over the 30°x30° region used here. This reduces the standard error in the mean AOD used in Figure 4 to values smaller than 0.002 for any month, which we now note in the text description. There is also the issue of representativeness of passive retrievals, such as MISR, MODIS, and VIIRS, as these are from clear sky only. This issue was directly addressed in Hong and Di Girolamo ((Hong and Di Girolamo, 2022)), where they used CALIPSO data to show that clear sky aerosol climatologies from passive sensors are also representative of all sky climatological conditions.  The manuscript was edited also as noted below to include notes on clear sky climatologies, clouds in the region, and MISR AOD as discussed.

1: "The overlapping of large fraction of cirrus clouds with lower clouds in the area (Hong and Di Girolamo, 2020) makes space-borne remote sensing of aerosol particles very challenging (Reid et al., 2013; Lin et al., 2014)."

2.1.1: "The AERONET observations were made during clear sky conditions, which has been shown (Hong and Di Girolamo, 2022) to be able to represent all sky conditions."

2.1.4: "Monthly 500 nm AOD data (Level 3 Global Aerosol: $0.5° \times 0.5°$ spatial resolution) from 2009 to 2018 are used from the Multi-angle Imaging SpectroRadiometer (MISR), (Diner et al., 2007; Garay et al., 2018) as regional (Southeast Asia) baseline remote sensing data to support the Manila Observatory AERONET data. The regional MISR data was used to confirm regional sources of aerosols that may be influencing the AOD over Metro Manila."

2.1.4: "Monthly mean AOD values were then calculated for each 0.5° grid point and then for the $30° \times 30°$ region, where the standard error in the monthly mean for the region is less than 0.002."

Lines 170-177: How were the NAAPS data products used? Where they used quantitatively or qualitatively?

Response: The maps of AOD and Surface Aerosol Concentrations for Southeast Asia were used qualitatively. This information is now more clearly indicated in the text (first paragraph of 2.1.5): "Archived maps of total and speciated optical depths and surface concentrations of sulfate, dust, and smoke for Southeast Asia are used from the Navy Aerosol Analysis and Prediction System (NAAPS: $1° \times 1°$ spatial resolution) (Lynch et al., 2016), and which are publicly available at https://www.nrlmry.navy.mil/aerosol/."

An additional sentence was added to the end of the section: "These maps help associate possible regional emission sources to extreme aerosol loading events in Manila Observatory."

Lines 178-181: Later discussion indicates use of other Worldview products. Please specify all products or images used.

Response: MODIS (Aqua and Terra) maps over Metro Manila and Southeast Asia were downloaded from Worldview. This information has been updated in the text." Archived maps of cloud fraction (Aqua MODIS and Terra MODIS) over Metro Manila and Southeast Asia were downloaded from NASA Worldview (*https://worldview.earthdata.nasa.gov)* for events of interest based on AERONET data."

Lines 184-185: The AERONET VSD is retrieved for discrete particle sizes and do not represent "bins" as mentioned here.

Response: The word "bins" has been changed to "discrete points". This is from 3.1 in https://aeronet.gsfc.nasa.gov/new_web/Documents/Inversion_products_for_V3.pdf.

Lines 193-194:  The dust category can include mixed aerosols as well as dust.  Lower FMF (<0.4 or <0.3) is more appropriate.  Also, the desert dust AOD may not retain high aerosol loading over 1000s of kms.  What wavelength is used for AOD, AE, and FMF in the table?  For example, the dust case identified over Manila for March 24-25, 2018, does not reach 0.3 at Level 2.0 AOD 500nm.  Which FMF wavelength is used? Please state in the caption. Overall, these threshold values for AOD, AE, and FMF are only estimates as they are not rigorous cutoffs for these air mass types and this should be discussed in the text referencing Table 2.

Response: This is true about the dust category being mixed with other aerosols, in the manuscript the air mass is renamed to mixed dust. The FMF values in Table 2 for the specific categories (clean marine, urban/industrial, biomass burning, and dust) were removed and the angstrom exponent values from literature (Kaskaoutis et al., 2009) were used. The wavelengths in Table 2 were indicated for those which had information about them, for example other studies just indicated the source of the data (MODIS).

The note about the lower AOD loading due to distance from the source has been added to the text in 3.4.2. "The AOD from both AERONET and MERRA-2 (Fig. 10b) are lower than 0.3 (the AOD threshold for dust in other studies, Table 2) because of the long distance from the source (thousands of kilometers)."

The text has been updated to indicate that these threshold values are estimates and that they can help in understanding aerosol sources as well as identify extreme cases. "The first four mentioned air mass types in Table 2 are the most general, four more classifications based on aerosol particle sources are included. The urban/industrial air mass type here refers to local combustion along with long-range transported biomass burning (Kaskaoutis et al., 2009). While these classifications are not rigid definitions of air masses, they help in understanding the sources that contribute to aerosols in Metro Manila and in identifying cases where certain sources are more influential than others."

Lines 214-219:  When was the dust case identified?  NAAPS itself is a model so how can it "confirm" the existence of dust over Manila?  Also, HYSPLIT uses reanalysis data and it too depends on a model and a number of assumptions in which is provides a possible transport pathway for aerosols.  Do you have surface based measurements confirming the dust reached Manila?

Response: The date for this case is 24 March 2018. The VSD for this case had a more dominant coarse contribution, which is the main reason for associating this as a dust case. We agree that these other sources of information (NAAPS and HYSPLIT) are not a ground truth to confirm the presence of an aerosol feature and are just supportive at most. In the first case study (Sect 2.3.1) we modified the text to be softer (changed "identify" to "provide support for" to reflect this reviewer comment: "were used to **provide support for** the likely source and transport pathway for the smoke cases".

Lines 224-225:  How were the NASA Worldview images used for verification?  For example, NASA Worldview provides many products.  Authors, need to use caution and understand the uncertainties related to MERRA-2 in regards to data assimilation, modeling, and determination of aerosol species. The MERRA-2 data set should not be treated as a measurement.

Response: The sentence was improved to state more clearly that it was MODIS and not MERRA-2 that was used to "verify" clouds: "The presence of clouds was verified qualitatively with MODIS (Aqua and Terra) imagery from NASA Worldview in the path of air parcels reaching Metro Manila based on HYSPLIT back-trajectories."

Lines 252-253: Where is the plot?

Response: The maps described here are in Figure 12 (b to d). The sentence was edited. "From the linear regression equation, the regression coefficient per grid was calculated. Each grid on the Southeast Asia map was colored based on the calculated regression coefficient value."

Line 278-279: AERONET data at Manila are considerably under-sampled during the months during the Summer Southwest Monsoon between May and October. For example, some years during this period, very few data were collected (e.g., 2013) due to the weather and the changing of the instrument. State in the text or in the figure the total number of observations used for each monthly average.

Response: The following was inserted into the main text: "(from available AERONET data of 513 points in August, 4015 points in February, and 5049 points in March)". The counts per month were also inserted into the figure caption. "(h) precipitable water (data counts per month Jan: 2131, Feb: 4015, Mar: 5049, Apr: 5844, May: 3448, Jun: 1696, Jul: 652, Aug: 513, Sep: 753, Oct: 1700, Nov: 2084, Dec: 1449);"

Line 284-288: Which level are temperature, relative humidity? It should be indicated that low-cloud fraction is from MERRA-2 with cloud top pressure > 680hPa. How is the distinction made between precipitating and non-precipitating clouds? It is very likely aerosols are washed out in precipitating clouds so partitioning by precipitating and non-precipitating clouds is important in the cloud processing assessment. Also, the cloud processing is difficult to determine monthly as these processes occur on the sub day temporal grid.

Response: Both temperature and relative humidity values are for 975 mb. This value has been indicated in the results text and the figure caption. "(a) temperature at 975 mb, (b) relative humidity at 975 mb,"

The text "(cloud top pressure > 680 hPa)" was inserted in the figure caption. There is no distinction made between precipitating and non-precipitating clouds in this study, however.

For this study we identified possible cloud processing for one date based on the AERONET volume size distribution and MERRA-2 data.

Line 308: What are the values in the legend and why is the scale so small (i.e., 10^-7)?

Response: The values in the legend are the density value contributions to calculated cumulative probability distribution surfaces, where the red areas have the greatest number of trajectories within a 100 km radius. The text of the caption was edited as follows. The legend was not included to avoid confusion.

"Density plots of HYSPLIT trajectories reaching Manila Observatory per month from 2009 to 2018. Red denotes areas with the greatest number of back trajectories within a 100 km radius. The colors represent density value contributions to Matlab-calculated cumulative probability distribution surfaces (100 km radius) from coordinates of three-day back trajectories of the specific months."

Line 334-342 (Figure 3): In Figure 3c, specify that these data are the SDA retrievals. What are the total number of observations and/or days for each monthly averages presented? The total number may explain some of the variations in the plots due to low sampling of AERONET data and quality controls. The Figures 3d and 3e are not clear on which data whisker plots refer to which wavelengths; please specify them on the plot. In Figure 3h, the blue line the ratio of RI between 440nm and 675nm but the red line is the ratio of RI between 440nm to the RI average of 670-1020nm. It seems either the wavelength should be 670nm or 675 consistently through the document.

Response: FMF has been specified as an SDA retrieval. "(c) spectral de-convolution algorithm (SDA) retrievals of fine mode fraction (FMF, 500nm)"

The total numbers per month have been indicated in the figure caption." Monthly characteristics of AERONET aerosol particle parameters: (a) aerosol optical depth (AOD, 500nm) with counts (Jan: 2107, Feb: 3931, Mar: 4923, Apr: 5755, May: 3389, Jun: 1653, Jul: 637, Aug: 483, Sep: 718, Oct: 1555, Nov: 2001, Dec: 1386) (b) extinction angstrom exponent (EAE, 440-870 nm) with counts (Jan: 102, Feb: 248, Mar: 312, Apr: 309, May: 137, Jun: 53, Jul: 14, Aug: 18, Sep: 18, Oct: 79, Nov: 77, Dec: 52), (c) spectral de-convolution algorithm (SDA) retrievals of fine mode fraction (FMF, 500nm) with the same counts as AOD, (d) single scattering albedo (SSA) from 440 nm (leftmost boxplot) to 1020 nm (rightmost boxplot) with counts (Jan: 6, Feb: 31, Mar: 62, Apr: 50, May: 29, Jun: 8, Aug: 3, Sep: 5, Oct: 17, Dec: 3), (e) asymmetry factor (AF) from 440 nm (leftmost boxplot) to 1020 nm (rightmost boxplot) with the same counts as EAE, (f) real and (g) imaginary refractive index (RI) values (440 nm) with the same counts as SSA,"

The wavelengths have been specified in the caption for Figures 3d and 3e as indicated in the text in the previous sentence.

The wavelength should be 675 nm in the caption for Figure 3h. This has been updated.

Line 350-351- MISR monthly averages are based on more limited data per month due to its orbit and measurement technique. Also, MISR over pass is in the afternoon so these data are biased to the afternoon clouds. Does MERRA-2 have the same constraint? Therefore, is MISR the appropriate instrument to compare monthly AOD? Why not use MODIS, VIIRS, or possibly Himawari AOD?

Response: The reviewer is incorrect: MISR is in a morning orbit (10:30 AM ECT descending branch), not an afternoon orbit. The morning orbit has a significant advantage over the afternoon orbit because cirrus coverage is at a minimum, thus allowing for more valid retrievals to occur. This, in part, is why we chose MISR. But there are other great reasons. MISR and MODIS AOD have extensive validation, more than any other passive AOD product. Unlike MODIS, MISR also includes an assessment of the impact of small cumulus (typical of the CAMP2Ex domain) contamination on aerosol retrievals, which was shown to have a negligible impact on MISR AOD and particle properties (Zhao et al., 2009). MISR also compares better with AERONET than MODIS, as cited in the original manuscript. MISR also provides particle size and particle shape segregated AOD, which no other passive sensor provides. As noted earlier, sampling noise for the large domain, 10-year monthly mean value is small relative to the seasonal signal that we are examining with the data. So, the choice to use MISR was logical. Not using other satellite data does not invalidate our use of MISR. The MERRA-2 reanalysis AOD product includes data assimilation from MODIS (Terra: 10:30 AM, Aqua: 1:30 PM), AVHRR (~ 1:30 AM, 9:30 AM, 1:30 PM, 9:30 PM), and MISR (10:30 AM) and are therefore constrained by these sensors. Studies in Asia (Xiao et al., 2009; Qi et al., 2013; Choi et al., 2019) show relatively higher MODIS AOD compared to MISR AOD and which could affect the MERRA-2 AOD. The subsection on MISR regional AOD was moved to 2.1.4 after all the datasets (i.e., MODIS is 2.1.2) that were used primarily for Metro Manila.

2.1.2: "The aerosol reanalysis data includes data assimilation of AOD from the Moderate Resolution Imaging Spectroradiometer (MODIS: Terra, 2000 to present and Aqua, 2002 to present), Advanced Very High Resolution Radiometer (AVHRR, 1979-2002), and Multiangle Imaging SpectroRadiometer (MISR, 2000-2014) (Buchard et al., 2017; Rizza et al., 2019)."

2.1.4: "Monthly 500 nm AOD data (Level 3 Global Aerosol: 0.5° × 0.5° spatial resolution) from 2009 to 2018 are used from the Multi-angle Imaging SpectroRadiometer (MISR), (Diner et al., 2007; Garay et al., 2018) as regional (Southeast Asia) baseline remote sensing data to support the Manila Observatory AERONET data. The regional MISR data was used to confirm regional sources of aerosols that may be influencing the AOD over Metro Manila. Level 3 MISR products are global maps of parameters available in Level 2 (measurements derived from the instrument data) products. MISR is ideal for remote sensing in the CAMP$^2$Ex region because it has an overpass at 10:30 AM ECT (descending mode) (when cirrus is minimal) and its retrievals have been shown to be unimpacted by small cumulus (Zhao et al., 2009), which are typical in the region. MISR has relatively more accurate AOD and agrees better with AERONET data compared to other satellite products due to its multi-angle measurements (Choi et al., 2019; Kuttippurath and Raj, 2021). The MISR sampling noise is relatively small due to the large domain and seasonal averages that are considered in this study. MISR is also the only passive sensor that speciates aerosol particle size and shape. All these factors led to the choice of using regional MISR data to associate long-range sources influencing AERONET data in Manila Observatory."

3.2.1: "Regional AOD (550 nm) over the larger Southeast Asia domain from MISR and MERRA-2 (Fig. 4) had a similarly large peak around the same time beginning in September until October which, however, was second only in magnitude to a March peak, which is influenced by biomass burning in Peninsular Southeast Asia (PSEA) (Gautam et al., 2013; Hyer et al., 2013;

Dong and Fu, 2015; Wang et al., 2015; Yang et al., 2022). This is consistent with the peak in speciated AOD due to fine (radii <0.7 µm), spherical, and absorbing aerosols that were observed by MISR from March to April (Fig. S1)."

3.2.1: "This dip was also observed in the regional AOD data (MISR and MERRA-2, Fig. 4). This is most probably due to the decrease in the AOD contribution from fine (radii <0.7 µm) and spherical particles based on size speciated MISR AOD (Fig. S1). Larger and non-spherical particle contributions to AOD increase in November in the Southeast Asia region. The MERRA-2 AOD is relatively higher than the MISR AOD probably due to assimilation of MODIS data into MERRA-2. Studies in Asia (Xiao et al., 2009; Qi et al., 2013; Choi et al., 2019) have observed relatively higher MODIS AOD compared to MISR AOD."

3.2.2: "The high EAE over Manila Observatory from July to September is probably regional in nature based on the MISR data showing increased EAE with increased AOD from fine, spherical, and absorptive particles (Fig. S1) in Southeast Asia during the same months."

3.2.2" "The lowest EAE values (0.08) and thus the largest particles were observed in December, which again may be regional in nature with MISR EAE also lowest during this time with increased AOD from larger and non-spherical particles (Fig. S1)."

Line 360-362: "0 for particles as large as cloud drops" – the desert dust aerosols can also generate EAE near and below zero. This statement "except for when the coarse mode has a large impact on the angstrom exponent" is not clear and needs revision.

Response: A reference for desert dust and sea salt has been added. And the statement referred to in the second sentence of this comment has been removed. "The EAE is usually greater for smaller particles (> 2 for small particles, ~4 for very small particles that undergo Rayleigh scattering, < 1 for large particles like sea salt and dust, and 0 for particles as large as cloud drops) (Schuster et al., 2006; Bergstrom et al., 2007)."

Line 371: EAE values less than 0.8 EAE are less likely to be fine mode dominated and/or impacted by optically thin cirrus clouds (especially in the Southeast Asia/Philippines region).

Response: The text was edited to indicate that most of the EAE values were greater than 0.8. "Of the high loading cases (AOD >1), the EAE values were mostly greater than 0.8 indicating fine mode particles (Che et al., 2015)."

Lines 458-461: However, Sinyuk et al. 2020 (https://amt.copernicus.org/articles/13/3375/2020/) showed that the AERONET real part of the refractive index is correlated to the fine mode size distribution contribution which makes is a less robust parameter.

Response: The following text was included in the mentioned section. "For the case of the AERONET data, which include refractive index values that are insensitive to coarse particles (Sinyuk et al., 2020), the focus of the discussion will be for fine mode particles and may be limited when coarse particles are involved."

Line 465-466:  The Authors have presented SSA at four wavelengths from AERONET and not just one.  AERONET SSA is computed using the four standard wavelengths (440, 670, 870, 1020nm) (e.g., Dubovik and King, 2000, Dubovik, et al. 2002).

Response: Thanks for pointing this out. We revised the text to not make this incorrect connection between wavelengths and SSA: "For this study, we examine refractive index values at 440 nm wavelength."

Line 545 (Figure 5):  Very interesting plot showing the variations of the size distributions from AERONET at Manila.  Can you please provide the related VSD properties (effective radius, median radius, standard deviation, peak radius of volume concentration) for each mode (fine and coarse) in a table or plot in the manuscript or supplementary?  Also, what is the average AOD and FMF related to each VSD?

Response: The VSD properties and AOD and FMF values were tabulated and added to the supplementary section. The table is shown below. The text was edited as well to refer to Table 1. "The fine mode median values peaked in the accumulation mode at 0.148 µm particle radius while the coarse mode median values peaked at 3.857 µm (Fig. 5a and Table S1).  The median coarse mode amplitudes and volume concentrations were higher than the fine mode amplitudes and volume concentrations for most of the year (DJF, MAM, and SON, Fig. 5b and Table S1), except during the southwest monsoon (JJA) when the fine mode amplitude and volume concentration was higher."

**Table S1:** Median aerosol properties associated with the volume size distribution data for the total, fine, and coarse fractions. The volume concentration ($C_v$) has units of $\mu m^3 \, \mu m^{-2}$. The following parameters have units of µm: radius at peak volume concentration ($r_{peak}$), effective radius ($r_{eff}$), volume mean radius ($r_v$), and standard deviation ($\sigma$).

| Time Frame | AOD | $C_v$ | FMF | Fine AOD | Fine $C_v$ | Fine $r_{peak}$ | Fine $r_{eff}$ | Fine $r_v$ | Fine $\sigma$ | Coarse AOD | Coarse $C_v$ | Coarse $r_{peak}$ | Coarse $r_{eff}$ | Coarse $r_v$ | Coarse $\sigma$ |
|---|---|---|---|---|---|---|---|---|---|---|---|---|---|---|---|
| ALL | 0.1674 | 0.0610 | 0.6514 | 0.1086 | 0.0220 | 0.1482 | 0.1450 | 0.1630 | 0.4890 | 0.0524 | 0.0360 | 3.8575 | 1.9690 | 2.6410 | 0.7330 |
| DJF | 0.1507 | 0.0590 | 0.6215 | 0.0907 | 0.0190 | 0.1482 | 0.1475 | 0.1640 | 0.4960 | 0.0557 | 0.0390 | 3.8575 | 2.0410 | 2.7300 | 0.7335 |
| MAM | 0.1791 | 0.0630 | 0.6774 | 0.1240 | 0.0235 | 0.1482 | 0.1430 | 0.1600 | 0.4820 | 0.0522 | 0.0360 | 3.8575 | 1.9435 | 2.5995 | 0.7330 |
| JJA | 0.1708 | 0.0520 | 0.7700 | 0.1400 | 0.0310 | 0.1482 | 0.1460 | 0.1670 | 0.5080 | 0.0360 | 0.0220 | 2.9400 | 2.1680 | 2.7520 | 0.6670 |
| SON | 0.1479 | 0.0575 | 0.5733 | 0.0869 | 0.0170 | 0.1482 | 0.1480 | 0.1665 | 0.5035 | 0.0570 | 0.0370 | 2.9400 | 1.9290 | 2.5640 | 0.7505 |

| | | | | | | | | | | | | | | | |
|---|---|---|---|---|---|---|---|---|---|---|---|---|---|---|---|
| AM | 0.1654 | 0.0600 | 0.6443 | 0.1067 | 0.0210 | 0.1482 | 0.1460 | 0.1640 | 0.4950 | 0.0518 | 0.0350 | 3.8575 | 1.9494 | 2.5925 | 0.7360 |
| PM | 0.1850 | 0.0575 | 0.6847 | 0.1264 | 0.0260 | 0.1482 | 0.1410 | 0.1550 | 0.4590 | 0.0555 | 0.0390 | 3.8575 | 2.2070 | 2.8850 | 0.7190 |

Lines 551-570:  The discussion of the VSD peaks needs further clarification.  For example, instead of "The average coarse mode peak (0.04 um3/um2) was the highest…," you could say, "The average coarse mode peak for the volume concentration (0.04 um3/um2) was the highest…"  Several other sentences in this paragraph are similarly vague and should be revised.

Response: The words "volume concentration" were added whenever the VSD magnitude and peaks were mentioned. An example is noted below:

3.2.6: "The median coarse mode amplitudes and volume concentrations were higher than the fine mode amplitudes and volume concentrations for most of the year (DJF, MAM, and SON, Fig. 5b and Table S1), except during the southwest monsoon (JJA) when the fine mode amplitude and volume concentration was higher."

Line 597-600:  What are the total number of AERONET measurements in each cluster? Figure 6 indicates that a total of 1345 VSDs were used for the cluster averaging.  How were the corresponding parameters correlated to each of these clusters?  Previous plots show mainly monthly averages however individual VSD retrievals and even VSD clusters are not explicitly tied to a month.  How are the corresponding parameters in Figure 7 grouped into the clusters?  The SDA FMF is not within inversion product so what timing threshold is used to link to the inversion data?  Also, the AOD 500nm is measured at a different time than the retrieval unless you are taking the AE and computing AOD 500nm between 440nm and 675nm.

Response: The number of AERONET measurements per cluster and the corresponding MERRA-2 data is the same as in Figure 6 (total of 1419). Figure 7 caption has been edited to include the counts. "Average compositional contributions to aerosol optical depth (AOD) from MERRA-2 per identified cluster (counts per cluster from 1 to 5: 830, 284, 166, 74, 65)." The corresponding AERONET and MERRA-2 parameters that were closest in time to the VSD date and time were considered for the plots in Figure 7. All the corresponding data were plotted as boxplots in Figure 7. We just associated the AOD 500 nm, EAE, and FMF data that was available that were closest in date and time to the VSD data date and time.

Line 668: Figure 8b, how well does the total AOD from MERRA-2 compare to AERONET AOD at the same time?  What wavelength is used for MERRA-2 data in the plot?  Please indicate this information in Figure 8b.  Also, if this is 500nm AOD then it appears underestimated at the specified times.  Figure 8d back trajectory analysis only shows for three days so it is difficult to determine the source region.  The Back trajectory analysis should be used between 5 and 7 days to better show the source region.

Response: The total AOD from MERRA-2 (550 nm) is slightly smaller for all the cases compared to the AERONET AOD (500 nm), except for the dust case (almost the same as the AERONET AOD at 500 nm). For comparison, the AOD from AERONET has been plotted alongside the compositional AOD from MERRA-2. The back trajectory analysis has been extended to 7 days for all the cases noted. The updated plot and caption are found below.

[Figure]

**Figure 8:** Case study of long-range transport (smoke – East Asia) around 1 April 2010. (a) AERONET VSDs at (blue) 00:01 and (red) 00:26 UTC, (b) AOD from AERONET (gray: median AOD  at 500 nm) and MERRA-2 hourly (green: black carbon, violet: dust, yellow: sea salt, orange: organic carbon, blue: sulfate) compositional contributions to AOD (550 nm) closest in time to 00:01 UTC, (c) NAAPS maps of total and compositional hourly AOD (orange/red: sulfate, green/yellow: dust, blue: smoke) and  sulfate, dust, and smoke surface concentrations at 00:00 UTC, and (d) HYSPLIT seven-day back-trajectories arriving at Manila Observatory at 00:00 UTC.

Line 697: Same as Figure 8 comments.

Response: Figure 9 and its captions have been updated as noted in the response to comments above for Figure 8.

Line 716: Same as Figure 8 comments.

Response: Figure 10 and its captions have been updated as noted in the response to comments above for Figure 8.

Line 741: Same as Figure 8 comments.

Response: Figure 11 and its captions have been updated as noted in the response to comments above for Figure 8.

Line 748: Section 3.6 seems out of place and perhaps should be part of Section 2.4, 2.5, or new 2.6.

Response: We thought hard about this but still think its current placement is suitable for how we feel the information can be presented.

[revised manuscript text omitted]

Response: We thank the two reviewers for the thoughtful suggestions and constructive criticism that have helped us to continue to improve our manuscript. Below we provide responses to referee comments and suggestions in blue font. All changes to the manuscript can be identified in the latest version submitted as well as in the marked-up manuscript at the end of this document.

Referee Comment: "An Aerosol Climatology via Remote Sensing over Metro Manila, Philippines"
Anonymous Referee #2
May 23, 2023

Lorenzo et al. have explored aerosol climatology over Manila using AERONET database. Besides, MISR AOD was considered with MERRA-2 data and other met. data to explain aerosol type and aerosol movement through wind. The manuscript in most cases lost direction and lack critical analysis, as too many datasets were used without much detail interpretation. Writing of the text also needs improvement.

General comments:

1. The manuscript lack of novelty. Aerosol climatology has not been constituted with long term database. Authors have explored too many aerosols dataset without being conclusive on any of these. Determination of aerosol type lacks more analysis and improvement of hypothesis.

   Response: Thank you for the critical feedback as it is helpful to see one side of a perspective on our work. The other side is that the analysis and results help to better understand aerosol characteristics in a climate-sensitive region of the world where aerosol measurements are very difficult to do. The AERONET database is in fact the longest continuous aerosol column database in the Philippines to date. We feel strongly that our analysis sheds important light on aerosol characteristics in a way that has not been done before, with the foundation being the AERONET dataset whose long-term nature has yet to be exploited in the way we did in this study. We have analyzed aerosol parameters besides AOD: which was the focus of other recent studies on columnar aerosol in the Philippines and in Metro Manila. The comparison of the Metro Manila data to regional Southeast Asia AOD shows a unique characteristic of Metro Manila that is affected by the West Pacific as well as the monsoon trough, that is not necessarily a Southeast Asian region-wide phenomena. Therefore we believe there is sufficient importance and new forms of results that are presented to continue the progress in understanding aerosol particle behavior in Metro Manila. As you will be able to see throughout the manuscript (a paragraph was added to the introduction as noted in the response to Specific Comment #2 below), we have made more revisions to hopefully drill home the point that the work is important to archive for the research community.

2. Significant part of case studies is based on NAAPS model outcome which is only used for regulatory forecast purposes and has uncertainty in model forecast over the Philippines.

Response: We used NAAPS maps to support the case studies which were based on the clustering results of AERONET VSD data. MERRA-2 was also used along with HYSPLIT back trajectories. The text was edited to show that the primary dataset used was AERONET (specifically VSD clustering results), with the other independent sources providing support. Reanalysis products such as MERRA-2 and NAAPS are quite helpful for conditions in which clouds affect remote sensing of aerosol particles such as in southeast Asia. NAAPS has been used in the way we did for a number of other studies aiming to have a supplementary source of support for air pollutant sources.

2.3 Extreme Event Analysis "Aerosol particle events based on the three clusters with the highest VSD concentrations were identified to characterize different types of sources and processes impacting aerosol particle columnar properties above Metro Manila. The three events are described below."

2.3.1. Smoke Long Range Transport "Events related to transported biomass burning/smoke were chosen from the AERONET VSD data that were clustered as urban/industrial (with a dominant submicrometer peak) (Eck et al., 1999) over Metro Manila. Cases with the highest black carbon contribution to total AOD from the MERRA-2 dataset were considered. Maps from NAAPS of high smoke contributions to AOD and surface smoke contributions in the direction of back-trajectories from the National Oceanic and Atmospheric Administration's (NOAA) Hybrid Single-Particle Lagrangian Integrated Trajectory (HYSPLIT) model (Stein et al., 2015; Rolph et al., 2017) were used to provide support for the likely source and transport pathway for the smoke cases."

2.3.2 Dust Long Range Transport "A dust transport case over Metro Manila was identified from the AERONET VSD dust cluster (with an enhanced coarse peak in the AERONET VSD compared to the submicrometer fraction) (Eck et al., 1999), the highest dust contribution to AOD from the MERRA-2 dataset, and high dust contributions to AOD from NAAPS. Surface dust concentrations from NAAPS along the HYSPLIT back-trajectories improved the plausibility of dust for this case."

3. Besides, there are some claims that need to be reverified by authors. For instance, availability of AOD dataset and related research over Southeast Asia (SEA) are plenty. SEA is one of the extensively explored regions of the world for aerosols because of its climate significance. However, in the Line 46, authors claim non availability of research on aerosols over Southeast Asia. This is certainly not true.

Response: We have revised the text in question to better articulate what the pressing needs are and to of course give credit to the significant volume of work done already: Introduction. "Although Southeast Asia is one of the most rapidly developing regions in the world with a growing number of extensive research conducted (Reid et al., 2023), there remain knowledge gaps related to aerosol particles in the area (Tsay et al., 2013; Lee et al., 2018; Chen et al., 2020; Amnuaylojaroen, 2023)."

4. Again, August is pointed to be the highest biomass burning month for Insular SEA. However, in fact, Sept.-Oct. is primarily reported to have widespread forest and peatland fires, burned over large parts of maritime southeast Asia, most notably in Indonesia, southern Sumatra and southern Kalimantan. In many reports, the increase in AOD in the Philippines during Oct. is a direct consequence of fire in neighboring Indonesia and corresponding wind movement in Sept. to Oct.

Response: The months of August to October were the months with the highest AERONET AOD median values. The text in the abstract noting only August as the month with highest biomass burning emissions was edited as follows. The text in 3.2.1 was edited to include "Indonesia" and references were added to the sentence. Other references were added to include studies showing AOD increase in the Philippines due to the Indonesia fires.

Abstract (L28): "Aerosol optical depth (AOD) values were highest from August to October, partly from fine urban aerosol particles, including soot, coinciding with the burning season in maritime Southeast Asia when smoke is often transported to Metro Manila during the southwest monsoon."

3.2.1: "This is the same time of year when biomass burning activities occur in the Indonesian region southwest of Metro Manila (Glover and Jessup, 1998; Kiely et al., 2019; Cahyono et al., 2022). Studies have shown that AOD in the Philippines increases during the biomass burning season in Indonesia (Nguyen et al., 2019; Caido et al., 2022)."

Specific comment:

1. There are many contradictory sentences explaining the results. Like in L28, highest AOD in August is contradictory to L325 and L329 with AOD 0.23 in October.

   Response: The highest AERONET AOD in Manila Observatory was from August (0.21) to September (0.23). The sentence (L28) was edited accordingly. MISR and MERRA-2 data represent regional AOD and show a peak at the same time beginning in September to October. The sentence (L325 and L329) was edited as well.

   Abstract (L28): "Aerosol optical depth (AOD) values were highest from August to October, partly from fine urban aerosol particles, including soot, coinciding with the burning season in Insular Southeast Asia when smoke is often transported to Metro Manila during the southwest monsoon."

   3.2.1: "Regional AOD (550 nm) over the larger Southeast Asia domain from MISR and MERRA-2 (Fig. 4) had a similarly large peak around the same time beginning in September until October which, however, was second only in magnitude to a March peak, which is influenced by biomass burning in Peninsular Southeast Asia (PSEA) (Gautam et al., 2013; Hyer et al., 2013)."

2. Introduction needs to be re-drafted. First paragraph does not conclude anything specific besides mentioning some previous research on the Philippines. Please include the findings on aerosols/aod/fire/trend from previous experiments. Content in the second paragraph is more relevant in the study area, section 2. Even the context of conducting this research is very briefly mentioned. Please emphasize on what new science questions can be answered by innovative analyses using combinations of data sets. Just because aerosol climatology has not been reported from a region using a new data product does not hold the novelty of the research.

   Response: We have revised the introduction to emphasize more findings (in the text below) from past experiments and to move some content (previously paragraph 2) to Section 2. With regard to the concern over a lack of new science questions, we list four science questions at the end of the Introduction Section.

   "Most of the past studies involving long-term remotely sensed aerosol particle data in Southeast Asia (Cohen, 2014; Nakata et al., 2018; Nguyen et al., 2019) had no specific focus on the Philippines. The Philippines is considered as part of the Maritime Continent (MC), the island nations sub-region of Southeast Asia. The other Southeast Asia sub-region, Peninsular Southeast Asia (PSEA), comprises those nations within the continental Asia land mass. These two regions have separate aerosol sources and climate, where MC is dependent on the intertropical convergent zone (ITCZ) and PSEA is dependent on both the ITCZ and monsoon systems (Dong and Fu, 2015). Only the southern part of the Philippines is climatologically part of MC (Ramage, 1971), however, and northwest Philippines, where Metro Manila is located, is affected by the monsoons and tropical cyclones aside from the ITCZ (Chang et al., 2005; Yumul Jr et al., 2010; Bagtasa, 2017). These unique meteorological influences and extensive local aerosol particle sources warrant a unique aerosol climatology over Metro Manila, one of a polluted source in a tropical marine environment, and its effects on cloud formation in the area. Aerosol effects on clouds in the marine environment are associated with the largest uncertainties in climate change research (Hendrickson et al., 2021; Wall et al., 2022) and the Philippines was ranked as the 5th country globally as most at risk to climate change and extreme weather from 1997 to 2018 (Eckstein et al., 2018). There have been several surface measurements of aerosol particles made in Metro Manila for the past 20 years (Oanh et al., 2006; Bautista VII et al., 2014; Cruz et al., 2019) but column-based ground-based measurements there are just beginning to be established (Dorado et al., 2001; Cruz et al., 2023; Ong et al., 2016). The AERONET sun photometer is one of the first long-term column-based aerosol instruments in Metro Manila and the Philippines (Ong et al., 2016)."

3. Drafting the manuscript in many cases is not appropriate. In L 88, authors emphasized that only AERONET data has been used to study whereas in table 1, it says for AERONET, MERRA-2 and PERSIANN. No discussion on MISR monthly AOD is included.

Response: The other complementary datasets used were also enumerated in the text (including MISR, HYSPLIT, NAAPS, and MODIS). Table 1 was updated to include the datasets specifically over Metro Manila, the other datasets over Southeast Asia like MISR, HYSPLIT, and NAAPS were discussed in sections 2.1.3, 2.1.4, and 2.1.5.

L88: "The goal of this study is to use multi-year AERONET data in Manila Observatory along with other complementary datasets (MERRA-2, PERSIANN, MISR, HYSPLIT, NAAPS, and MODIS) to address the following questions:"

**2. Methods:** "This work relies on analysis of several datasets over Metro Manila and regional Southeast Asia summarized in Table 1 and in the following subsections. The common time range used for all datasets is between January 2009 and October 2018."

**Table 1:** Summary of datasets over Metro Manila used in this work covering the period from January 2009 to October 2018.

| Parameter | Data Source | Spatial Coverage | Time Coverage |
| --- | --- | --- | --- |
| Aerosol Optical Depth (500 nm) | AERONET | 14.635°N, 121.078°E | Jan 2009 - Oct 2018 |
| Asymmetry Factor (440 nm - 1020 nm) | AERONET | 14.635°N, 121.078°E | Jan 2009 - Oct 2018 |
| Extinction Angstrom Exponent (440 nm -870 nm) | AERONET | 14.635°N, 121.078°E | Jan 2009 - Oct 2018 |
| Fine Mode Fraction | AERONET | 14.635°N, 121.078°E | Jan 2009 - Oct 2018 |
| Precipitable Water | AERONET | 14.635°N, 121.078°E | Jan 2009 - Oct 2018 |
| Single Scattering Albedo (440 nm - 1020 nm) | AERONET | 14.635°N, 121.078°E | Jan 2009 - Oct 2018 |
| Refractive Index (Real and Imaginary; 440 nm - 1020 nm) | AERONET | 14.635°N, 121.078°E | Jan 2009 - Oct 2018 |
| Volume Size Distribution | AERONET | 14.635°N, 121.078°E | Jan 2009 - Oct 2018 |
| Low Cloud Fraction (MODIS) | MERRA-2 | 14.3°N - 14.8°N, 120.75°E - 121.25°E | Jan 2009 - Dec 2018 |
| Planetary Boundary Layer Height | MERRA-2 | 14.3°N - 14.8°N, 120.75°E - 121.25°E | Jan 2009 - Dec 2018 |
| Relative Humidity (975 mb) | MERRA-2 | 14.3°N - 14.8°N, 120.75°E - 121.25°E | Jan 2009 - Dec 2018 |
| Sea Level Pressure | MERRA-2 | 14.3°N - 14.8°N, 120.75°E - 121.25°E | Jan 2009 - Dec 2018 |
| Temperature (975 mb) | MERRA-2 | 14.3°N - 14.8°N, 120.75°E - 121.25°E | Jan 2009 - Dec 2018 |
| Wind (975 mb) | MERRA-2 | 14.3°N - 14.8°N, 120.75°E - 121.25°E | Jan 2009 - Dec 2018 |
| Total Extinction Aerosol Optical Depth (550 nm) | MERRA-2 | 14.3°N - 14.8°N, 120.75°E - 121.25°E | Jan 2009 - Dec 2018 |
| Sulfate, Black Carbon, Organic Carbon, Dust, and Sea Salt Extinction Aerosol Optical Depth (550 nm) | MERRA-2 | 14.3°N - 14.8°N, 120.75°E - 121.25°E | Jan 2009 - Dec 2018 |
| Precipitation | PERSIANN | 14.3°N - 14.8°N, 120.75°E - 121.25°E | Jan 2009 - Dec 2018 |

4. 2.1.3 What was the purpose of comparing monthly MISR 0.5x0.5 data against AERONET and MERRA- AOD? This does not conclude anything scientific on aerosol climatology.

Response: The AERONET data over Metro Manila is influenced by both local and long-range aerosol particle sources, and the only way to assess the influence of these long-range sources is to look at a large (i.e. 30° × 30°) region around Manila, which is why we looked at both the MISR and MERRA-2 regional data. Regional AOD values from MISR (remote sensing) and MERRA-2 (reanalysis) were used as independent sources of support for the long-range aerosol particles seen over Metro Manila AOD from AERONET. Supporting monthly regional AOD from independent instruments (AERONET and MISR remote sensors) and methods (MERRA-2 reanalysis which includes data assimilation from AOD from MODIS and MISR) helped to reinforce regional sources of aerosol particles that influence the Metro Manila air column. MISR also has aerosol particle shape and size speciated AOD, which provide further support for the other AERONET aerosol parameters. A figure was added to show the shape and size speciated AOD (Fig. S1). The size speciated data helps reinforce the regional influence on the aerosol particles over Metro Manila especially during high AOD times from July to September (when high AERONET EAE associated with fine, spherical, and absorptive particles based on MISR data) and times with the low AOD in December (large particles based on MISR size speciated AOD). Edits in the manuscript to reflect these responses are found below.

[Figure]

Figure S1: MISR monthly mean time series of 550 nm AOD (total, large (particle radii > 0.7 µm), medium (particle radii from 0.35 to 0.7 µm), small (particle radii < 0.35 µm), non-spherical spherical, and absorption optical depth) and angstrom exponent (AE) for March 2000 to December 2020 for 116.5°E to 128.5°E; 6.5°N to 22.5°N.

2.1.2 "The total MERRA-2 AOD for the region was used along with MISR AOD data to assess the influence of long-range sources to the aerosol column over Manila Observatory."

2.1.4 "Monthly 500 nm AOD data (Level 3 Global Aerosol: 0.5° × 0.5° spatial resolution) from 2009 to 2018 are used from the Multi-angle Imaging SpectroRadiometer (MISR), (Diner et al., 2007; Garay et al., 2018) as regional (Southeast Asia) baseline remote sensing data to support the Manila Observatory AERONET data. The regional MISR data was used to confirm regional sources of aerosols that may be influencing the AOD over Metro Manila."

2.1.4 "MISR is ideal for remote sensing in the CAMP2Ex region because it has an overpass at 10:30 AM ECT (descending mode) (when cirrus is minimal) and its retrievals have been shown to be unimpacted by small cumulus (Zhao et al., 2009), which are typical in the region. MISR has relatively more accurate AOD and agrees better with AERONET data compared to other satellite products due to its multi-angle measurements (Choi et al., 2019; Kuttippurath and Raj, 2021). The MISR sampling noise is relatively small due to the large domain and seasonal averages that are considered in this study. MISR is also the only passive sensor that speciates aerosol particle size and shape. All these factors led to the choice of using regional MISR data to associate long-range sources influencing AERONET data in Manila Observatory."

3.2.1 "Regional AOD (550 nm) over the larger Southeast Asia domain from MISR and MERRA-2 (Fig. 4) had a similarly large peak from September to October which, however, was second only in magnitude to a March peak, which is influenced by biomass burning in Peninsular Southeast Asia (PSEA) (Gautam et al., 2013; Hyer et al., 2013). This is consistent with the peak in speciated AOD due to fine (radii <0.7 µm), spherical, and absorbing aerosols that were observed by MISR from March to April (Fig. S1).: 3.2.1 "This dip was also observed in the regional AOD data (MISR and MERRA-2, Fig. 4). This is most probably due to the decrease in the AOD contribution from fine (radii <0.7 µm) and spherical particles based on size speciated MISR AOD (Fig. S1). Larger and non-spherical particle contributions to AOD increase in November in the Southeast Asia region. The MERRA-2 AOD is relatively higher than the MISR AOD probably due to assimilation of MODIS data into MERRA-2. Studies in Asia (Xiao et al., 2009; Qi et al., 2013; Choi et al., 2019) have observed relatively higher MODIS AOD compared to MISR AOD."

3.2.2 "The high EAE over Manila Observatory from July to September is probably regional in nature based on the MISR data showing increased EAE with increased AOD from fine, spherical, and absorptive particles (Fig. S1) in Southeast Asia during the same months."

3.2.2 "The lowest EAE values (0.08) and thus the largest particles were observed in December, which again may be regional in nature with MISR EAE also lowest during this time with increased AOD from larger and non-spherical particles (Fig. S1)."

5. The criteria for considering Table 2 has some ambiguity. Please explain what was the basis of considering FMF to sort fine and coarse aerosols? Again for a country like the Philippines with a very low annual AOD (~0.2), how accurate is it to separate marine and industrial aerosols based on FMF?

Response: The table's criteria has been used in other regions with even lower AOD. The references are indicated in the table. An updated table, including more angstrom exponent threshold values, is shown in response #12. The table's criteria represents one possible way to do size classification (i.e. FMF has been used in other studies (Kleidman et al., 2005)) as a proxy for remotely sensed observations. We have also added a sentence at the end of the paragraph preceding the table to contextualize our use of the thresholds.

"While these classifications are not rigid definitions of air masses, they help in understanding the sources that contribute to aerosols in Metro Manila and in identifying cases where certain sources are more influential than others."

6. Table 2: Its also strange that no biomass burning aerosols were considered as air mass type when biomass burning is an important contributor. Beside authors are working on smoke aerosol transport in section 2.3.1 while no such classification was made in Table 2. Please justify.

Response: The long-range transport of biomass burning aerosol particles was considered as part of the urban/industrial air mass type which includes both local combustion and long-range transported biomass burning. The text in section 2.2 has been updated to include this information. Table 2 has also been updated to include a threshold for biomass burning based on literature. The changes to the texts are indicated below. The changes to the table are in response #12.

2.2 "The urban/industrial air mass type here refers to local combustion along with long-range transported biomass burning (Kaskaoutis et al., 2009)."

3.3.2 "Combining this and results from the previous sections confirms that cluster 4 can be an urban/industrial source given that it had the highest median accumulated mode peak and organic carbon contribution to total AOD among the clusters. The cluster 4 air mass is probably from local sources and transported biomass burning emissions. The high median EAE (1.40, Fig. 7c) may be associated with aerosol particles due to biomass burning (Deep et al., 2021).

7. The NAAPS model outputs are not always convincing enough to detect regional emission sources. NAAPS model aerosol forecasts are available on a 1°× 1° grid. However, use of NAAPS model forecast over Philippines is questionable as 'number of AOT assimilations available in and around the Philippines is limited because of the pervasive cloud cover, making model outputs of AOT subject to uncertainty for this region (https://acp.copernicus.org/articles/22/12961/2022/)'.

Response: We use NAAPS to provide support for the AERONET data. We are not using it to detect the regional sources but just associate possible regional sources during extreme aerosol loading events based on AERONET data in Manila Observatory. We edited the text to note these points (including the resolution of NAAPS AOT reanalysis product).

2.1.5 NAAPS: "Archived maps of total and speciated optical depths and surface concentrations of sulfate, dust, and smoke for Southeast Asia are used from the Navy Aerosol Analysis and Prediction System (NAAPS: $1° \times 1°$ spatial resolution) (Lynch et al., 2016), and which are publicly available at https://www.nrlmry.navy.mil/aerosol/." "These maps help associate possible regional emission sources to extreme aerosol loading events in Manila Observatory."

2.3.1 Smoke Long Range Transport: "Maps of surface smoke contributions from NAAPS as well as back-trajectories from the National Oceanic and Atmospheric Administration's (NOAA) Hybrid Single-Particle Lagrangian Integrated Trajectory (HYSPLIT) model (Stein et al., 2015; Rolph et al., 2017) were used to provide support for the likely source and transport pathway for the smoke cases."

8. Section 2.4: Why not EOF was performed on monthly mean MISR AOD instead of a reanalysis product?

   Response: EOF needs a data set with no data gaps, and the 10-year monthly MISR AOD for Southeast Asia had geographical data gaps at certain times. The sentence in section 2.4 was edited as follows.

   "EOF analysis needs a complete dataset with no data gaps, which is not available with pure satellite retrievals like MISR; MERRA-2 reanalysis data alleviate this issue."

9. Fig. 1 & 2 is not required in the main text, move it to the supplementary file. Reduce the related discussion on meteorology as this paper is not focused on aerosol -meteorology interaction but the aerosol climatology. Entire section 3.1 should be removed/deleted keeping Fig. 1 and Fig. 2 in supplementary.

   Response: We appreciate this comment in that it provides a perspective other than ours to let us think more deeply about the way we originally presented our results. After much more thought, we still feel Figures 1-2 and associated Section 3.1 are important to retain to provide context for the study and to allow it to be intercompared with related studies from other regions.

10. Fig. 3a: trend in AOD is not clear, make adjustments in y axis.

Response: The y-axis has been adjusted. The following text has been edited in the caption.

[Figure]

"Figure 3: Monthly characteristics of AERONET aerosol particle parameters: (a) aerosol optical depth (AOD, 500nm with y-axis until 1.0 only for larger boxplot resolution) with counts (Jan: 2107, Feb: 3931, Mar: 4923, Apr: 5755, May: 3389, Jun: 1653, Jul: 637, Aug: 483, Sep: 718, Oct: 1555, Nov: 2001, Dec: 1386)."

11. L 330: The major contributor of biomass burning emission in peninsular Southeast Asia is emission from Indonesia with much higher fire spots in October compared to that of Philippines in March. It's highly unusual to have a greater AOD peak in March compared to October. Justify.

Response: Biomass burning in Peninsular Southeast Asia (what is also referred to as Indochina Peninsula, (Dong and Fu, 2015)) peaks in March. Previous studies based on AERONET and MODIS data show peak AOD over Peninsular Southeast Asia in March (Gautam et al., 2013; Wang et al., 2015), MERRA-2 biomass burning emissions over Peninsular Southeast Asia peak in March as well (Yang et al., 2022). The text has been edited to include Dong and Fu, Wang et al., and Yang et al. as references. The approximate distance from Peninsular Southeast Asia to the Philippines is also included in the text.

"Regional AOD (550 nm) over the larger Southeast Asia domain from MISR and MERRA-2 (Fig. 4) had a similarly large peak around the same time beginning in September until October which, however, was second only in magnitude to a March peak, which is influenced by biomass burning in Peninsular Southeast Asia (PSEA) (Gautam et al., 2013; Hyer et al., 2013; Dong and Fu, 2015; Wang et al., 2015; Yang et al., 2022). This is consistent with the peak in speciated AOD due to fine (radii <0.7 µm), spherical, and absorbing aerosols that were observed by MISR from March to April (Fig. S1). This larger peak in March, attributed to PSEA (which is ~2000 km west of the Philippines), was not as prevalent in the AERONET AOD data over Manila Observatory in Metro Manila due to the dominant easterly winds in the Philippines in March (Fig. 2c) and more localized sources."

12. 3.2.3: Author should add SSA to identify aerosol mass type as in Table 2. Besides avoid explaining individual aerosol optical properties in the result discussion part, instead focus on results. This is true for all sub sections in 3.2.

Response: SSA has been added as criteria in Table 2 as shown below. The text under SSA has been edited to include the comparison to biomass burning SSA observed values as indicated in Table 2. We maintained the flow of discussion for ease of reading and discussion.

**Table 2:** Summary of threshold values of aerosol optical depth (AOD), angstrom exponent (AE), fine mode fraction (FMF), and single scattering albedo (SSA) used to identify air mass types.

| Air Mass Type | AOD | AE | FMF | SSA | Source |
|---|---|---|---|---|---|
| Clean Fine | < 0.1[a] | > 1[a] | > 0.7[a] | - | Sorooshian et al., 2013 |
| Polluted Fine | > 0.1[a] | > 1[a] | > 0.7[a] | - | Sorooshian et al., 2013 |
| Clean Coarse | < 0.1[a] | < 1[a] | < 0.3[a] | - | Sorooshian et al., 2013 |
| Polluted Coarse | > 0.1[a] | < 1[a] | < 0.3[a] | - | Sorooshian et al., 2013 |
| Clean Marine | < 0.2[b] | < 0.9[d] | - | 0.98[e] | Kaskaoutis et al., 2009 Dubovik et al., 2002 |
| Urban/Industrial | > 0.2[b] | > 1[d] | - | 0.9-0.98[e] | Kaskaoutis et al., 2009 Dubovik et al., 2002 |
| Biomass Burning | - | > 1.4[a] | - | 0.89-0.95[e] | Deep et al., 2021 Dubovik et al., 2002 |
| Desert Dust | > 0.3[c] | < 1[d] | - | 0.92-0.93[e] | Kaskaoutis et al., 2009 Deep et al., 2021 Dubovik et al., 2002 |

[a] from MODIS     [c] AOD at 400 nm     [e] SSA at 440 nm

[b] AOD at 500 nm     [d] AE at 380 nm to 870 nm

3.2.3 "Monthly median SSA values were largest in June (0.94 at 440 nm), suggesting the presence of more reflective aerosol particles, and smallest in August (0.88 at 440 nm and 0.78 at 1020 nm) suggesting more absorptive particles that are similar in range to the SSA of biomass burning particles (Table 2)."

13. Section 3.3.2: Include SSA as an additional parameter to characterize aerosol mass and re discuss the result.

Response: SSA has been added to the figure referenced in section 3.3.2. The updated figure is shown below. The discussions in the text have also been updated to include SSA and are noted below as well.

[Figure]

**Figure 7:** (a) Average compositional contributions to aerosol optical depth (AOD, 550 nm) from MERRA-2 per identified cluster (counts per cluster from 1 to 5: 830, 284, 166, 74, 65). Boxplots of AERONET (b) total AOD (500 nm), (c) single scattering albedo (SSA, 440 nm), (d) extinction angstrom exponent (EAE, 440 nm – 870 nm total), and (e) fine mode fraction (FMF, 500 nm) per cluster.

[revised manuscript text omitted]

---

## Author Response (AR2)

We thank the anonymous reviewer for going over our responses and edits to the manuscript. We affirm that we tried our best to respond to the anonymous reviewer's concerns. The most recent points of Anonymous Referee # 2 (AR2) are below (in black), and we have indicated our responses in blue. We refer to sections of the previous Author's Response document (EGUSPHERE on 13 July 2023) to help us address some of the concerns in this current response document. Many references are made to our last set of responses, which we feel may have been overlooked or misinterpreted.

The submitted manuscript explored aerosol climatology over Manila, Philippines. I find several hypotheses considered by authors and justifications provided in the first draft were not updated in revision. Many of the claims in the revision were not in line with evidence provided by contemporary researchers. I am pointing out few of the major arguments of the authors which are actually not true, primarily vague. Based on the author's response and submitted revision, I recommend the article to be rejected.

We responded to all of the comments of AR2 (pages 16 to 32 of the Author's Response on 13 Jul 2023) and indicated how we edited the manuscript. That 16-page response to AR2 was both complete and accurate. In cases where we felt our discussion (Sections 3.1 and 3.2) was important in the flow of our study, we explained to AR2 our reasoning in the response to Specific Comment # 9 (page 24) and Specific Comment # 12 (SC # 12, page 26).

AR2 in this latest set of reviews states that "claims in the revision were not in line with evidence provided by contemporary researchers" with no specifics on which claims are in question. AR2 also doesn't cite any references to the evidence that they are not "in line with". This makes it difficult for us to address this comment. So that we can address this comment, we would be grateful for clarification with regard to the claims and references which we are not "in line with".

1. Major concern is the scientific novelty and lack of scientific questions that are addressed by the authors. The manuscript is mere of a report of observations made by authors during a certain period. Entire section 3.2 is mere reporting of observation without much scientific context.

    Although it may be that the reviewer does not agree, the manuscript reports on both novel observations and analysis. We go beyond reporting of mere observations and do a detailed analysis and interpretation of the AERONET data with supporting data. Our work also presents interpretation of these data at local (Metro Manila) and regional (30 degree lat/lon centered on Metro Manila) scales, including sources, meteorology, and aerosol characteristics (AOD, EAE, FMF, SSA, AF, and RI). Unfortunately, the reviewer did not provide specifics for us to address. For example, which other publications report all of our findings? We think we did a thorough review based on the 157 references provided in the manuscript, but if we missed something, we would like to know. We note that the depth and order of our discussion in 3.2 builds up our analysis of aerosol

characteristics enabling a fuller understanding of aerosol monthly behavior (beyond AOD) from 2009 to 2018. This analysis is novel as AERONET is the first long-term ground-based aerosol columnar measurement system that was set up in the Metro Manila in 2009. We also performed cluster analysis on the volume size distribution from 2009 to 2018 supported by data from MERRA-2. We have already noted in Specific Comment # 2 (page 19) why our study is important. Our science questions are at the end of the Introduction as noted also in Specific Comment # 2 (page 19).

2. Determination of aerosol type lacks science. SSA was included as a matrix to identify aerosol subtypes but was not actually used properly to distinguish aerosol types. Some of the aerosol types were based on FMF, some on AE, AOD value >0.1 only to indicate polluted AOD. There are several ambiguities in selecting aerosol properties in identifying aerosol types.

Our response to Specific Comment # 13 (pages 28 to 29) addresses the comment above. We also note that we are not determining "aerosol type", as AR2 suggests, but rather clusters with similar airmass aerosol characteristics. SSA and the other aerosol parameters were used as criteria for air mass assignments for identified clusters as noted in our response in SC # 13. Our approach in using SSA, EAE, FMF, and AOD for airmass aerosol classification is dependent on available thresholds from previous studies and has been used in many other parts of the world for airmass aerosol classification, with example citations already provided in our manuscript: "Dubovik et al., 2002; Pace et al., 2006; Kaskaoutis et al., 2007; Kaskaoutis et al., 2009; Sorooshian et al., 2013; Kumar et al., 2014; Sharma et al., 2014; Che et al., 2015; Kumar et al., 2015; Deep et al., 2021".

3. Figure 1 and 4 does not conclude anything. Why compare MISR against MERRA 2? What does it prove?

The analysis of the AERONET aerosol parameters depended on the data that was in Figure 1. We referenced it 13 times in 3.1 and 7 times in 3.2. We could not have as complete an analysis as we did without considering the meteorology and water vapor over Metro Manila that we had based on MERRA-2, PERSIANN, and AERONET. We can move Figure 1 to the Supplementary section, although we feel this is a detriment to the paper (e.g., 20 references made to the figure in the paper) and the other reviewer supported its inclusion.

We had a thorough response to AR2 about using MISR and MERRA-2 in our Author's Response document (Specific Comment # 4, pages 20-23): "Regional AOD values from MISR (remote sensing) and MERRA-2 (reanalysis) were used as independent sources of support for the long-range aerosol particles seen over Metro Manila AOD from AERONET." Both MISR and MERRA-2 average monthly AOD (2009 – 2018 for the 30° x 30° region) peak in March, which proves that there is a regional peak in AOD in Southeast Asia in March that is not as evident in the AERONET AOD over Metro Manila. MISR has additional information on the size, shape, and absorptivity of particles

that can give clues about the source of the regional AOD peak in March (fine, spherical, and absorbing particles).

4.  Significant part of case studies is based on NAAPS model outcome which does not provide much detail on actual aerosol climatology. Again, the authors explain "We use NAAPS to provide support for the AERONET data". These maps help associate possible regional emission sources to extreme aerosol loading events in Manila Observatory". In fact, use of NAAPS model forecast over Philippines is questionable as number of AOT assimilations available in and around the Philippines is limited because of the pervasive cloud cover.

NAAPS was used qualitatively in the analysis of case studies that were associated with the identified airmass aerosol clusters from AERONET data. Reanalysis products such as MERRA-2 and NAAPS help in conditions in which clouds affect remote sensing of aerosol particles such as in Southeast Asia. NAAPS has been used in the way we did for a large number of other published studies (a few of which we cited in the edited manuscript as noted below) aiming to have a supplementary source of support for air pollutant sources. The very purpose of reanalysis data is to fill in the 4-D space of meteorological and pollution conditions in the best way possible, while still recognizing its limitations – which we did in our manuscript.

Additional text after the last sentence of 2.1.5: "Previous studies have used NAAPS data for an overview of aerosol sources in specific regions of interest (Ross et al., 2018; Foth et al., 2019; Markowicz et al., 2021; Harenda et al., 2022; Mims III, 2022). More recent studies show the need to improve aerosol representation in NAAPS (Edwards et al., 2022), so we will use NAAPS qualitatively, together with MERRA-2 compositional AOD data and back-trajectories, for an overview of aerosol sources that may contribute to extreme events with high AOD from AERONET."

5.  What was the purpose of comparing monthly MISR 0.5x0.5 data against AERONET and MERRA- AOD ? This proves nothing.

We note the following edits in the manuscript that we made in Specific Comment # 4 (page 22, 2.1.2) and added the actual total region (30° x 30°) over which the data was averaged in to the manuscript (2.1.2): "The total MERRA-2 AOD (mean over 30° x 30° region) for the region was used along with MISR AOD data (mean over 30° x 30° region) to assess the influence of long-range sources to the aerosol column over Manila Observatory." The AOD peak in March (MISR and MERRA-2) proves that there is a regional peak in AOD in Southeast Asia in March that is not observed over Metro Manila (where there is no distinct AOD peak in March). The speciated MISR AOD data helps reinforce the regional influence on the aerosol particles over Metro Manila especially during high AOD times from July to September (fine, spherical, and absorptive particles) that are consistent with AERONET data.

6. How MISR 0.5x0.5 AOD data was considered as regional (Southeast Asia) baseline remote sensing data to support the Manila Observatory AERONET data.

Average monthly AOD values from the 30° × 30° region (0.25°N – 30.25°N and 104.75°E – 134.75°E) from 2009 to 2018 are used from MISR. The bounding coordinates are included in the text in 2.1.4 for clarification.

"Monthly 500 nm AOD data (Level 3 Global Aerosol: 0.5° × 0.5° spatial resolution in the region 0.25°N – 30.25°N and 104.75°E – 134.75°E) from 2009 to 2018 are used from the Multi-angle Imaging SpectroRadiometer (MISR), (Diner et al., 2007; 222 Garay et al., 2018) as regional (Southeast Asia) baseline remote sensing data to support the Manila Observatory AERONET data."

7. "The high EAE over Manila Observatory from July to September is probably regional in nature based on the MISR data" This is no scientific evidence against this claim.

The text has been edited in 3.2.2 to communicate what is meant by the authors.

"The high EAE over Manila Observatory from July to September is consistent with the regional (30° latitude x 30°longitude) MISR data that shows increased AOD from fine, spherical, and absorptive particles (Fig. S1) in Southeast Asia during the same months. This suggests that the high EAE observed at the Manila Observatory during these months is not necessarily from local sources."

Additional References:

Edwards, E.-L., Reid, J. S., Xian, P., Burton, S. P., Cook, A. L., Crosbie, E. C., Fenn, M. A., Ferrare, R. A., Freeman, S. W., and Hair, J. W.: Assessment of NAAPS-RA performance in Maritime Southeast Asia during CAMP2Ex, Atmospheric Chemistry and Physics, 22, 12961-12983, 2022.
Foth, A., Kanitz, T., Engelmann, R., Baars, H., Radenz, M., Seifert, P., Barja, B., Fromm, M., Kalesse, H., and Ansmann, A.: Vertical aerosol distribution in the southern hemispheric midlatitudes as observed with lidar in Punta Arenas, Chile (53.2∘ S and 70.9∘ W), during ALPACA, Atmospheric Chemistry and Physics, 19, 6217-6233, 2019.
Harenda, K. M., Markowicz, K. M., Poczta, P., Stachlewska, I. S., Bojanowski, J. S., Czernecki, B., McArthur, A., Schuetemeyer, D., and Chojnicki, B. H.: Estimation of the effects of aerosol optical properties on peatland production in Rzecin, Poland, Agricultural and Forest Meteorology, 316, 108861, 2022.
Markowicz, K., Zawadzka-Manko, O., Lisok, J., Chilinski, M., and Xian, P.: The impact of moderately absorbing aerosol on surface sensible, latent, and net radiative fluxes during the summer of 2015 in Central Europe, Journal of Aerosol Science, 151, 105627, 2021.
Mims III, F. M.: A 30-Year Climatology (1990–2020) of Aerosol Optical Depth and Total Column Water Vapor and Ozone over Texas, Bulletin of the American Meteorological Society, 103, E101-E109, 2022.

Ross, A. D., Holz, R. E., Quinn, G., Reid, J. S., Xian, P., Turk, F. J., and Posselt, D. J.: Exploring the first aerosol indirect effect over Southeast Asia using a 10-year collocated MODIS, CALIOP, and model dataset, Atmospheric Chemistry and Physics, 18, 12747-12764, 2018.